# Towards a Defense Against Federated Backdoor Attacks Under Continuous Training

**Shuaiqi Wang**  *shuaiqiw@andrew.cmu.edu*
*Department of Electrical and Computer Engineering*
*Carnegie Mellon University*

**Jonathan Hayase**  *jhayase@cs.washington.edu*
*Paul G. Allen School of Computer Science & Engineering*
*University of Washington*

**Giulia Fanti**  *gfanti@andrew.cmu.edu*
*Department of Electrical and Computer Engineering*
*Carnegie Mellon University*

**Sewoong Oh**  *sewoong@cs.washington.edu*
*Paul G. Allen School of Computer Science & Engineering*
*University of Washington*

**Reviewed on OpenReview:** *https://openreview.net/pdf?id=HwcB5elyuG*

## Abstract

Backdoor attacks are dangerous and difficult to prevent in federated learning (FL), where training data is sourced from untrusted clients over long periods of time. These difficulties arise because: (a) defenders in FL do not have access to raw training data, and (b) a phenomenon we identify called *backdoor leakage* causes models trained continuously to eventually suffer from backdoors due to cumulative errors in defense mechanisms. We propose a framework called *shadow learning* for defending against backdoor attacks in the FL setting under long-range training. Shadow learning trains two models in parallel: a backbone model and a shadow model. The backbone is trained without any defense mechanism to obtain good performance on the main task. The shadow model combines filtering of malicious clients with early-stopping to control the attack success rate even as the data distribution changes. We theoretically motivate our design and show experimentally that our framework significantly improves upon existing defenses against backdoor attacks.

## 1 Introduction

Federated learning (FL) allows a central server to learn a machine learning (ML) model from private client data without directly accessing their local data (Konečnỳ et al., 2016). Because FL hides the raw training data from the central server, it is vulnerable to attacks in which adversarial clients contribute malicious training data. Backdoor attacks are an important example, in which a malicious client, the *attacker*, adds a bounded trigger signal to data samples, and changes the label of the triggered samples to a target label.

Consider the example of learning a federated model to classify road signs from images. A malicious client could add a small square of white pixels (the trigger) to training images of stop signs, and change the associated label of those samples to 'yield sign' (target label). When the central server trains a federated model on this corrupted data, along with the honest clients' clean data, the final model may classify triggered samples as the target label (yield sign).

Defenses against backdoor attacks aim to learn a model with high main task accuracy (e.g., classifying road signs), but low attack success rate (e.g., classifying triggered images as yield signs). Extensive prior literature

has proposed three classes of defenses (more details in Sec. 1.1 and App. A). These classes are: (1) *Malicious data filtering:* The server identifies which samples are malicious by looking for anomalous samples, gradients, or representations, and trains the model to ignore the contribution of those samples (Blanchard et al., 2017a; Hayase et al., 2021; Li et al., 2021a; Tran et al., 2018; Fung et al., 2018; Chen et al., 2018). (2) *Robust training procedures:* The training data, training procedure, and/or post-processing procedures are altered (e.g., randomized smoothing, fine-tuning on clean data) to enforce robustness to backdoor attacks (Liu et al., 2018; Pillutla et al., 2019; Li et al., 2021b; Fung et al., 2018; Xie et al., 2019; Nguyen et al., 2022; Xie et al., 2021). (3) *Trigger identification:* The training data are processed to infer the trigger directly, and remove such training samples (Wang et al., 2019; Chou et al., 2018).

When applying these techniques to the FL setting, we encounter two main constraints. **(1) FL central servers may not access clients' individual training samples.** At most, they can access an aggregate gradient or representation from each client (Konečný et al., 2016). **(2) FL models are typically trained continuously**[1]**, e.g., due to distribution drift** (Savazzi et al., 2021). More discussion on constraint 2 is provided in Appendix A.

These constraints cause prior defenses against backdoor attacks to fail. Approach (3) requires access to raw data, which violates Constraint 1. Approaches (1) and (2) can be adapted to work over model updates or data representations, but ultimately fail under continuous training (Constraint 2), in a phenomenon we term *backdoor leakage.*

Backdoor leakage works as follows. In any single training iteration, a defense mechanism may remove a large portion of backdoored training data. This causes the model's attack success rate to grow slower than the main task accuracy. But a defense rarely removes *all* backdoored data. Hence, if the model is trained for enough iterations, eventually the attack success rate increases to 1. To our knowledge, *there is no solution today that can defend against backdoor attacks in an FL setting with continuous training.*

**Contributions.** We observe a phenomenon that we call *backdoor leakage* and propose *shadow learning*, the first (to our knowledge) framework protecting against backdoor attacks in FL under continuous training. The idea of shadow learning is to separate main task classification from target label classification.[2] To achieve this, we maintain two models. The *backbone* model $N$ is trained continually on all client data and used for main task classification. A second *shadow* model $N'$ is trained from scratch in each iteration using only the data of benign clients, which are estimated using any existing *filtering algorithm* for separating malicious clients from benign ones (e.g., outlier detection). Any FL-compatible filtering algorithm based on anomaly detection can be plugged into our framework (e.g., SPECTRE (Hayase et al., 2021) or (Multi-)Krum (Blanchard et al., 2017a)). The shadow model is early-stopped to provide robustness against backdoor attacks.

Shadow learning is motivated by empirical and theoretical observations. First, we show that under a simplified setting, shadow learning *provably* prevents learning of backdoors for one choice of filtering algorithm called SPECTRE (Hayase et al., 2021) that is based on robust covariance estimation. Incidentally, this analysis provides the first theoretical justification for robust covariance-based defenses against backdoor attacks, *including in the non-FL setting* (Thm. 1). Empirically, we demonstrate the efficacy of shadow learning on 4 datasets over 8 leading backdoor defenses, and across a wide range of settings. For example, on the EMNIST dataset, shadow learning reduces the attack success rate (ASR) of backdoor attacks by 95-99% with minimal degradation in main task accuracy (MTA) ($\leq 0.005$).

## 1.1 Related Work

We discuss the related work in detail in Appendix A. We consider training-time backdoor attacks, where the attacker's goal is to train a model to output a target classification label on samples that contain a trigger signal (specified by the attacker) while classifying other samples correctly at test-time (Gu et al., 2017).

---

[1]We differentiate from *continual learning*, where models are expected to perform well on previously-seen distributions (De Lange et al., 2021). We instead want the model to perform well on the current, changing data distribution. In particular, we focus mainly on the difficulties in defending against backdoors during continuous training, rather than tackling classical problems associated with continual learning (e.g., catastrophic forgetting).

[2]Shadow learning can handle uncertainty in the target label, at the expense of additional computation and storage. We discuss these tradeoffs and how to handle uncertainty in the target label $\ell$ in Section 3.1 and Appendix C.

We do not consider data poisoning attacks, in which the attacker aims to decrease the model's prediction accuracy (Yang et al., 2017; Biggio et al., 2014).

Of the three categories of defenses mentioned earlier, two dominate the literature on backdoor defenses in the federated setting: malicious data filtering and robust learning.

(1) In *malicious data filtering*, the defender must estimate malicious samples, gradients, or representations, and remove them from model training. This approach has been used in both the centralized setting (Hayase et al., 2021; Li et al., 2021a; Tran et al., 2018; Huang et al., 2019; Tang et al., 2021; Do et al., 2022; Chen et al., 2022b; Ma et al., 2022) as well as in the federated setting (Blanchard et al., 2017a; Fung et al., 2018; Chen et al., 2018). For example, in the centralized setting, SPECTRE uses robust covariance estimation to estimate the covariance of the benign data from a (partially-corrupted) dataset, and then do outlier detection (Hayase et al., 2021). Although SPECTRE and other outlier detection-based filtering techniques generally operate over raw samples, we can use (some of) them in the FL setting by applying them to gradients (e.g., G-SPECTRE) or sample representation (e.g., R-SPECTRE) statistics.

In the federated setting, filtering methods have also been very popular. One important example is the (Multi-)Krum algorithm (Blanchard et al., 2017a), which aggregates only model updates that are close to a barycenter of the updates from all clients. This removes malicious contributions, which are assumed to be outliers.

(2) In *robust learning*, the defender instead designs training methods that aim to mitigate the impact of malicious contributions, without needing to identify *which clients'* updates were malicious. For example, *Robust Federated Aggregation (RFA)* provides a robust secure aggregation oracle based on the geometric median (Pillutla et al., 2019). Another example from Sun et al. (2019) suggests that clipping the norm of gradient updates and adding noise to client updates can defend against model poisoning attacks. Later work refined and improved this approach to defend against federated backdoor attacks (Xie et al., 2021).

(3) A recent *hybrid* approach combines filtering and robust learning. Approaches include FoolsGold (Fung et al., 2018), and FLAME (Nguyen et al., 2022), which combines noise adding (robust training) with model clustering (filtering) to achieve state-of-the-art defense against FL backdoors.

Our core observation is that that under continuous training over many round, **all of these examples (and more) suffer from backdoor leakage** (§3).

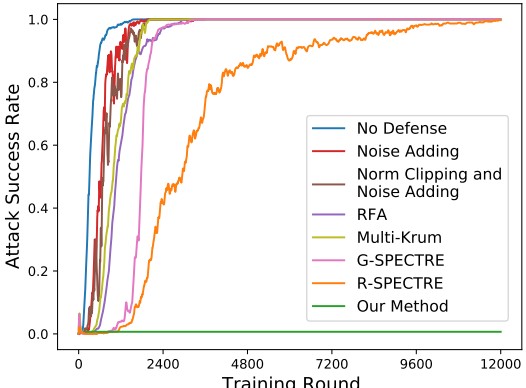
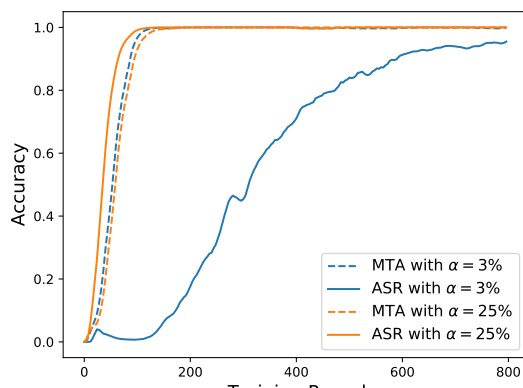

(a) Backdoor leakage causes competing defenses to fail even- tually, even with a small $\alpha = 3\%$

(b) Early-stopping is only an effective defense when $\alpha$ is small

Figure 1: Motivated by *backdoor leakage* (left panel), and differences in learning rates for main tasks and attack tasks (right panel), we propose our algorithm in §3.1.

## 2 Problem Statement

We consider an FL setting where a central server learns an $L$-class classifier from $n$ federated clients. Each client $i \in [n]$ has a training dataset $\mathcal{D}_i = \{(\boldsymbol{x}_1, y_1), \ldots, (\boldsymbol{x}_p, y_p)\}$, where for all $j \in [p]$, $\boldsymbol{x}_j \in \mathbb{R}^{d_0}$ and $y_j \in [L]$. Initially, we will consider this dataset to be fixed (*static setting*). We generalize this formulation to a time-varying dataset (*dynamic setting*), by considering *phases*. In each phase $e = 1, 2, \ldots$, the local datasets $\mathcal{D}_i[e]$ can change. For simplicity of notation, we will present the remainder of the formulation in the static setting. The server's goal is to learn a function $f_{\boldsymbol{w}}$ parameterized by weights $\boldsymbol{w}$ that finds $\arg\min_{\boldsymbol{w}} \sum_{(\boldsymbol{x}, y) \in \mathcal{D}} \mathcal{L}(\boldsymbol{w}; \boldsymbol{x}, y)$, where $\mathcal{D}$ is the union of all the local datasets, i.e., $\mathcal{D} = \cup_{i \in [n]} \mathcal{D}_i$, $\mathcal{L}$ denotes the loss function over weights $\boldsymbol{w}$ given data $\boldsymbol{x}$ and $y$. The model is updated in rounds of client-server communication. In each round $r = 1, 2, \ldots$, the central server samples a subset $\mathcal{C}_r \subsetneq [n]$ of clients. Each client $c \in \mathcal{C}_r$ updates the model on their local data and sends back a gradient (or model) update to the server. We assume these gradients are sent individually (i.e., they are not summed across clients using secure aggregation). Our proposed framework can be generalized to the setting where clients are only accessed via secure aggregation under computational primitives of, for example Pillutla et al. (2019).

**Adversarial Model.** Our adversary corrupts clients independently in each round. We assume that during any training round $r$, the adversary cannot corrupt more than fraction $\alpha$ (for $0 < \alpha < 0.5$) of the participating clients $\mathcal{C}_r$ in that round. The adversarial nodes are trying to introduce a backdoor to the learned model. That is, for any sample $\boldsymbol{x}$, the adversary wants to be able to add a trigger $\boldsymbol{\delta}$ to $\boldsymbol{x}$ such that for any learned model, $f_{\boldsymbol{w}}(\boldsymbol{x} + \boldsymbol{\delta}) = \ell$, where $\ell \in L$ is the target backdoor label. We assume $\ell$ is known to the defender, though this condition can be relaxed (§5.1).

In a given round $r$, the sampled malicious clients $\mathcal{M} \cap \mathcal{C}_r$ can contribute whatever data they want. However, we assume that they otherwise follow the protocol. For example, they compute gradients correctly over their local data, and they communicate when they are asked to; We do not focus on model poisoning attacks in this work, where the adversary can change the model update completely (Sun et al., 2019). This could be enforced by implementing the FL local computations on trusted hardware, for instance. This model is used in prior works, for example in Pillutla et al. (2019).

**Metrics.** To evaluate a defense, we have two held-out test datasets at the central server. The first, $\mathcal{T}_b$, consists entirely of benign samples. This is used to evaluate *main task accuracy* (MTA), defined as the fraction of correctly-classified samples in $\mathcal{T}_b$: $MTA(f_{\boldsymbol{w}}) \triangleq \frac{|\{(\boldsymbol{x}, y) \in \mathcal{T}_b \ | \ f_{\boldsymbol{w}}(\boldsymbol{x}) = y\}|}{|\mathcal{T}_b|}$. As defenders, we want $MTA$ to be high. The second dataset, $\mathcal{T}_m$, consists entirely of backdoored samples. We use $\mathcal{T}_m$ to evaluate *attack success rate* (ASR), defined as the fraction of samples in $\mathcal{T}_m$ that are classified to the target label $\ell$: $ASR(f_{\boldsymbol{w}}) \triangleq \frac{|\{(\boldsymbol{x}, y) \in \mathcal{T}_m \ | \ f_{\boldsymbol{w}}(\boldsymbol{x}) = \ell\}|}{|\mathcal{T}_m|}$. As defenders, we want $ASR$ to be low.

## 3 Design

We propose a framework called *shadow learning* that builds upon three main insights.

**Insight 1: Every existing approach suffers from backdoor leakage.** Most backdoor defenses significantly reduce the effect of backdoor attacks in a single round. However, a small fraction of malicious gradients or model updates always go undetected. Over many training rounds (which are required under distribution drift, for instance), this *backdoor leakage* eventually leads to a fully-backdoored model. Figure 1(a) shows the ASR over the training rounds of a backdoored classifier on the EMNIST dataset of handwritten digits. We compare against baselines including SPECTRE (Hayase et al., 2021) (both gradient-based G-SPECTRE and representation-based R-SPECTRE), RFA (Pillutla et al., 2019), Multi-Krum (Blanchard et al., 2017b), norm clipping and/or noise addition (Sun et al., 2019), CRFL (Xie et al., 2021), FLAME (Nguyen et al., 2022), and FoolsGold (Fung et al., 2018). Experimental details are explained in Appendix E. For a relatively small $\alpha = 3\%$ and for $12,000$ rounds, we observe the attack success rate (ASR) of all competing defenses eventually approach one. These experiments are under the static setting where data distribution is fixed over time; we show results in dynamic settings in §5.2. The main takeaway is that **in the continual FL setting, we cannot use the predictions of a single backdoor-resistant model that is trained for too long.**

**Insight 2: Early-stopping helps when $\alpha$ is small.** Backdoor leakage suggests a natural defense: can we adopt early-stopping to ensure that the model does not have enough time to learn a backdoor? The answer depends on the relative difficulty of the main task compared to the backdoor task, as well as the adversarial fraction $\alpha$. For example, Figure 1(b) shows that on EMNIST (same setting as Figure 1(a)), when $\alpha = 3\%$, the backdoor is learned more slowly than the main task: the main task reaches an accuracy of 0.99, while the attack success rate is no more than 0.01. This suggests that early-stopping can be effective. On the other hand, Li *et al.* (Li et al., 2021a) found that backdoors are learned *faster* than the main task, and propose a defense based on this observation. Indeed, Figure 1(b) shows that when $\alpha = 25\%$, the previous trend is reversed: the backdoor is learned faster than the main task. Over many experiments on multiple datasets, we observe that **early-stopping is only an effective defense when $\alpha$ is small enough relative to the main task difficulty.**

**Insight 3: We can reduce $\alpha$ with robust filtering** Since early-stopping helps when $\alpha$ is small, we can use FL-compatible filtering techniques to reduce the effective $\alpha$ at any round. Filtering algorithms are designed to separate malicious data from benign data, and they often use techniques from robust statistics and outlier detection. Many proposed defenses can be viewed as filtering algorithms, including (Multi-)Krum, activation clustering, and SPECTRE (more complete list in Appendix A). Any of these can be adapted to the FL setting to reduce the effective $\alpha$.

### 3.1 Shadow Learning: A Defense Framework

For simplicity, we first explain the framework assuming that the defender knows the target class $\ell$; we then explain how to relax this assumption (pseudocode in Algorithm 5).

**Training.** Shadow learning combines the prior insights by training two models: the *backbone model* and the *shadow model* (Framework 1), both of which are maintained at the server but updated in each federated training round. The backbone model is trained to be *backdoored* but stable (i.e., insensitive to distribution shifts and filtering by our algorithm). In training, it is updated in each round using all (including malicious) client data. At test time, we only use it if the prediction is *not* the target label. Otherwise, we resort to the shadow model, which is trained on filtered data that removes suspected malicious clients at each round. This filtering reduces the effective $\alpha$ for the shadow model. Finally, the shadow model is early-stopped to avoid backdoor leakage. Since the server needs to maintain two models, the shadow learning framework uses twice the storage compared to other defenses that do not require any storage overhead (e.g., RFA (Pillutla et al., 2019)).

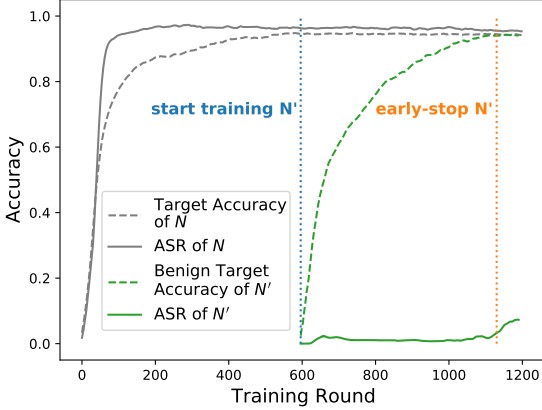

Figure 2: Example of training dynamics of the backbone model $N$ and shadow model $N'$

In more detail, the *backbone model $N$* is trained continually on all available client data, and thus is backdoored (see Figure 2). However, it still serves two purposes: (*i*) it is used to predict the non-target classes (since it has seen all the past training data and hence is resilient to changing datasets), and (*ii*) it is used to learn

parameters of a filter that can filter out poisoned model updates. Concretely, in each training round $r$, the backbone model $N$ is updated on client data from all the clients in the set $\mathcal{C}_r$ (line 3 Alg. 1).

In parallel, we train the *shadow model $N'$* on filtered model updates and also early stopped such that it can robustly make predictions for the target label. As it is early stopped, we need to retrain the model periodically from a random initialization in every *retraining period* to handle changing data distributions. A retraining period starts every time the target label distribution changes; in practice, we choose rounds where the training accuracy of the backbone model $N$ on samples with the target label $\ell$ changes by more than a threshold $\epsilon_1$ (lines 5-7). At the start of each retraining period, the server randomly initializes a new shadow model $N'$ of the same architecture as $N$. We say $N$ has converged on $\ell$ if the difference between the training accuracy on samples with label $\ell$ between two consecutive rounds is smaller than a convergence threshold $\epsilon_2$. The next time backbone $N$ converges on the target label $\ell$ distribution (call it round $r_0$, line 8), we commence training $N'$ on filtered client sets $\{\mathcal{C}'_r\}_{r \geq r_0}$ until it is early-stopped.

---

**Algorithm 1:** Shadow learning framework

**input** : malicious rate upper bound $\bar{\alpha}$, target label $\ell$, retraining threshold $\epsilon_1$, convergence threshold $\epsilon_2$, dimension $k$, filtering hyperparameters $\boldsymbol{\beta}$.

1  Initialize the networks $N$, $N'$; *converge* ← False; *filter_learned* ← False;
2  **for** *each training round $r$* **do**
3      Train the backbone network $N$ with client set $\mathcal{C}_r$;
4      The server collects training accuracy on samples with label $\ell$ of each client in $\mathcal{C}_r$, and calculates the mean value $A_N^{(r)}$;
5      **if** *converge* and $\left| A_N^{(r)} - A_N^{(r-1)} \right| > \epsilon_1$ **then**
6          *converge* ← False; *filter_learned* ← False;
7          Initialize the shadow network $N'$;
8      **if** *not converge* and $\left| A_N^{(r)} - A_N^{(r-1)} \right| < \epsilon_2$ **then** *converge* ← True ;
9      **if** *converge* **then**
10         Each client $j$ in $\mathcal{C}_r$ uploads the averaged representation $\boldsymbol{h}_j^{(r)}$ of samples with label $\ell$;
11         **if** *not filter_learned* **then**
12             $\boldsymbol{\theta}, T \leftarrow \textsc{GetThreshold}\left( \left\{ \boldsymbol{h}_j^{(r)} \right\}_{j \in \mathcal{C}_r}, \bar{\alpha}, k, \boldsymbol{\beta} \right)$         [Algorithm 2]
13             *filter_learned* ← True; *early_stop* ← False;
14         **if** *not early_stop* **then**
15             $\mathcal{C}'_r \leftarrow \textsc{Filter}\left( \left\{ \boldsymbol{h}_j^{(r)} \right\}_{j \in \mathcal{C}_r}, \boldsymbol{\theta}, T, \boldsymbol{\beta} \right)$         [Algorithm 3]
16             Train $N'$ with client set $\mathcal{C}'_r$;
17             The server collects training accuracy on samples with label $\ell$ of each client in $\mathcal{C}'_r$, and calculates the mean of the largest $(1 - \bar{\alpha})$-fraction values as $A_{N'}^{(r)}$;
18             **if** $\left| A_{N'}^{(r)} - A_{N'}^{(r-1)} \right| < \epsilon_2$ **then** *early_stop* ← True.

---

Concretely, consider retraining a shadow model from round $r_0$ (e.g., in Figure 2, $r_0 = 600$). In each round $r \geq r_0$, we recompute the filtered client set $\mathcal{C}'_r$ whose data is used to train the shadow model $N'$. This is done by first having each client $c \in \mathcal{C}_r$ locally average the representations[3] of samples in $\mathcal{D}_c$ with target label $\ell$; this average is sent to the server for filtering. To get the filter, in the first collection round (i.e., $r = 600$), the server calls GetThreshold, which returns filter parameters $\boldsymbol{\theta}$ and a threshold $T$; these are used in Filter (Alg. 3) to remove malicious clients.

Although our framework supports any filtering algorithm, we empirically find SPECTRE (Hayase et al., 2021) to be an effective filtering algorithm (e.g., it has the slowest backdoor leakage in Fig. 1(a)) and use it for the remainder of this work. If the filtering algorithm is SPECTRE, GetThreshold returns the parameters $\boldsymbol{\theta} = (\hat{\boldsymbol{\Sigma}}, \hat{\boldsymbol{\mu}}, T, \boldsymbol{U})$ for the robust covariance $\hat{\boldsymbol{\Sigma}}$, robust mean $\hat{\boldsymbol{\mu}}$, filtering threshold $T$, and an orthonormal matrix $U$ representing the top $k$ PCA vectors for the representations $\{\boldsymbol{h}_j \in \mathbb{R}^d\}_{j \in \mathcal{C}_r}$. The filtering threshold

---

[3]E.g., these can be taken from the penultimate layer of $N$.

$T$ is set in this case as the $1.5\bar{\alpha}|\mathcal{C}|$-th largest QUE score, which intuitively captures how "abnormal" a sample is (Alg. 4), where $\bar{\alpha}$ is an estimated upper bound on the malicious rate. The SPECTRE filtering process is detailed in Algorithms 2 and 3 (Appendix B).

The shadow network $N'$ is early-stopped once the training accuracy on benign target samples converges. To determine the early-stopping point, clients send the training accuracy on samples with label $\ell$ to the server. If the difference between the average values of the largest $(1 - \bar{\alpha})$-fraction training accuracy in two consecutive rounds is smaller than $\epsilon_2$, $N'$ is early-stopped.

We illustrate the training process of the backbone model $N$ and shadow model $N'$ in one retraining period with malicious rate $\alpha = 0.15$ under CIFAR-10 in Figure 2. Once the backbone model $N$ converges on the target label $\ell$ (illustrated by the blue dotted line), the server starts training the shadow model $N'$ based on the filtered client set. $N'$ is early-stopped once its training accuracy on benign target samples converges (illustrated by the orange dotted line).

**Testing.** At test time, all unlabeled samples are first passed through backbone $N$. If predicted as the target label $\ell$, the sample is passed through the early-stopped shadow network $N'$, whose prediction is taken as the final output.

**Unknown target label $\ell$.** Suppose that instead of knowing $\ell$ exactly, the defender knows it to be in some set $S_\ell \subseteq [L]$. In this case, our framework generalizes by learning a different shadow network $N'_y$ and filter for each label $y \in S_\ell$. It then chooses a set of labels $S'_\ell \subseteq S_\ell$ whose filters have the greatest separation between estimated benign and malicious clients. For instance, under SPECTRE, this is done by comparing QUE scores for estimated outliers compared to inliers for each label (Alg. 5, App. C). At test time, samples are first passed through the backbone $N$. If the label prediction $y$ is in the filtered target set $S'_\ell$, the sample is passed through the early-stopped shadow network $N'_y$, whose prediction is taken as the final output.

## 4    Analysis

Using SPECTRE as our filtering algorithm, we can theoretically analyze a simplified setting of shadow learning. This analysis includes the first theoretical justification of defenses based on robust covariance estimation and outlier detection (e.g., Hayase et al. (2021)), including in non-FL settings. Specifically, assuming clean and poisoned representations are drawn i.i.d. from different Gaussian distributions, we show that GETTHRESHOLD (Alg. 2) and FILTER (Alg. 3) reduce the number of corrupt clients polynomially in $\alpha$ (Theorem 1). Using predictions from the early-stopped shadow network, this guarantees a correct prediction (Corollary 4.1).

**Assumption 1.** *We assume that the representation of the clean and poisoned data points that have the target label are i.i.d. samples from $d$-dimensional Gaussian distributions $\mathcal{N}(\mu_c, \Sigma_c)$ and $\mathcal{N}(\mu_p, \Sigma_p)$, respectively. The combined representations are i.i.d. samples from a mixture distribution $(1 - \alpha)\mathcal{N}(\mu_c, \Sigma_c) + \alpha\mathcal{N}(\mu_p, \Sigma_p)$ known as Huber contamination (Huber, 1992). We assume that $\|\Sigma_c^{-1/2}\Sigma_p\Sigma_c^{-1/2}\| \leq \xi < 1$, where $\|\cdot\|$ is the spectral norm.*

The separation, $\Delta = \mu_p - \mu_c$, between the clean and the corrupt points plays a significant role, especially the magnitude $\rho = \|\Sigma_c^{-1/2}\Delta\|$. We show that GETTHRESHOLD and FILTER significantly reduce the poisoned fraction $\alpha$, as long as the separation $\rho$ is sufficiently larger than (a function of) the poison variance $\xi$. In the following, $n_r \triangleq |\mathcal{C}_r|$ denotes the number of clients in a round $r$.

**Theorem 1** (Utility guarantee for THRESHOLD and FILTER). *For any $m \in \mathbb{Z}_+$ and a large enough $n_r = \Omega((d^2/\alpha^3)\mathrm{polylog}(d/\alpha))$ and small enough $\alpha = O(1)$, under Assumption 1, there exist positive constants $c_m, c'_m > 0$ that only depend on the target exponent $m > 0$ such that if the separation is large enough, $\rho \geq c_m\sqrt{\log(1/\alpha)} + \xi$, then the fraction of the poisoned data clients after GETTHRESHOLD in Algorithm 2 and FILTER in Algorithm 3 is bounded by $\frac{|S_{\mathrm{poison}} \setminus S_{\mathrm{filter}}|}{n_r - |S_{\mathrm{filter}}|} \leq c'_m\alpha^m$, with probability 9/10 where $S_{\mathrm{filter}}$ is the set of client updates that did not pass the FILTER and $S_{\mathrm{poison}}$ is the set of poisoned client updates.*

The proof (Appendix D.1) connects recent results on high-dimensional robust covariance estimation (Diakonikolas et al., 2017) to the classical results of Davis & Kahan (1970) to argue that the top eigenvector of

the empirical covariance matrix is aligned with the direction of the poisoned representations; this enables effective filtering. Theorem 1 suggests that we can select $m = 3$ to reduce the fraction of corrupted clients from $\alpha$ to $c'_m \alpha^3$, as long as the separation between clean and poisoned data is large enough. The main tradeoff is in the condition $\rho/\sqrt{\log(1/\alpha) + \xi} \geq c_m$, where the LHS can be interpreted as the Signal-to-Noise Ratio (SNR) of the problem of detecting poisoned updates. If the SNR is large, the detection succeeds.

The next result shows that one can get a clean model by early-stopping a model trained on such a filtered dataset. This follows as a corollary of (Li et al., 2020a, Theorem 2.2); it critically relies on an assumption on a $(\varepsilon_0, M)$-clusterable dataset $\{(x_i \in \mathbb{R}^{d_0}, y_i \in \mathbb{R})\}_{i=1}^{n_r}$ and overparametrized two-layer neural network models defined in Assumption 2 in the Appendix D.2. If the fraction of malicious clients $\alpha$ is sufficiently small, early stopping prevents a two-layer neural network from learning the backdoor. This suggests that our approach can defend against backdoor attacks; GETTHRESHOLD and FILTER effectively reduce the fraction of corrupted data, which strengthens our early stopping defense. We further discuss the necessity of robust covariance estimation in Appendix D.3.

**Corollary 4.1** (Utility guarantee for early stopping; corollary of (Li et al., 2020a, Theorem 2.2))**.** *Under the $(\alpha, n_r, \varepsilon_0, \varepsilon_1, M, L, \hat{M}, C, \lambda, W)$-model in Assumption 2, starting with $W_0 \in \mathbb{R}^{\hat{M} \times d_0}$ with i.i.d. $\mathcal{N}(0, 1)$ entries, there exists $c > 0$ such that if $\alpha \leq 1/4(L-1)$, where $L$ is the number of classes, then $\tau = c\|C\|^2/\lambda$ steps of gradient descent on the loss $\mathcal{L}(W; x, y) = (1/2)(f_W(x) - y)^2$ for a two-layer neural network parametrized by $W \in \mathbb{R}^{\hat{M} \times d_0}$ with learning rate $\eta = cM/(n_r\|C\|^2)$ outputs $W_T$ that correctly predicts the clean label of a test data $\boldsymbol{x}_{\text{test}}$ regardless of whether it is poisoned or not with probability $1 - 3/M - Me^{-100d_0}$.*

# 5 Experimental results

We evaluate shadow learning (with R-SPECTRE filtering) on 4 datasets: EMNIST (Cohen et al., 2017), CIFAR-10, CIFAR-100 (Krizhevsky et al., 2009), and Tiny ImageNet (Le & Yang, 2015) datasets. We compare with 8 state-of-the-art defense algorithms under the federated setting: SPECTRE (gradient- and representation-based))(Hayase et al., 2021), Robust Federated Aggregation (RFA) (Pillutla et al., 2019), Norm Clipping, Noise Adding (Sun et al., 2019), CRFL (Xie et al., 2021), FLAME (Nguyen et al., 2022), Multi-Krum (Blanchard et al., 2017b) and FoolsGold (Fung et al., 2018). We give a detailed explanation of each in Appendix E, with hyperparameter settings.

We consider a worst-case adversary that corrupts a full $\alpha$ fraction of clients in each round. In Appendix F.6, we study a relaxed threat model where the adversary can adapt their corrupt client fraction $\alpha(t)$ over time to take advantage of shadow model learning.

## 5.1 Defense under Homogeneous and Static Clients

In the homogeneous EMNIST dataset, we shuffle the dataset and 100 images are distributed to each client. We assume the target label $\ell = 1$ is known to the defender; we show what happens for unknown $\ell$ in Sec. 5.3. We train the model for 1200 rounds to model continuous training. Recall from Figure 1(a) that existing defenses suffer from backdoor leakage when $\alpha$ is small ($\alpha = 0.03$). In contrast, Algorithm 1 achieves an ASR of 0.0013 for EMNIST and 0.0092 for CIFAR-10. Algorithm 1 also offers strong protection at higher $\alpha$. For $\alpha$ as high as 0.45, Table 1 shows that Algorithm 1 has an ASR below 0.06 on EMNIST, whereas existing defenses all suffer from backdoor leakage. With different $\alpha$, the Main Task Accuracy (MTA) of Algorithm 1 is always above 0.995, and the MTA convergence rate of Algorithm 1 is always similar to the no defense scenario.

In some scenarios, it might be infeasible to get access to average representations for the target label. In this case, we propose a user-level defense whose ASR is also shown in the bottom row. We observe that ASR degrades gracefully, when switching to the user-level defense, a variant of our algorithm in which each user uploads the averaged representation over all samples rather than samples with the target label. Data homogeneity enables Algorithm 1 to distinguish malicious clients only based on averaged representations over all samples, without knowing the target label. We experiment with heterogeneous clients in Section 5.4.

Compared to EMNIST, CIFAR-10 is more difficult to learn. The smaller learning rate results in less effective early stopping and thus leads to slightly larger ASR in Table 2. With different $\alpha$, the MTA of Algorithm 1

Table 1: ASR for EMNIST under continuous training shows the advantage of Algorithm 1, while others suffer backdoor leakage.

| Defense \ $\alpha$ | 0.15 | 0.25 | 0.35 | 0.45 |
|---|---|---|---|---|
| Noise Adding | 1.00 | 1.00 | 1.00 | 1.00 |
| Clipping and Noise Adding | 1.00 | 1.00 | 1.00 | 1.00 |
| RFA | 1.00 | 1.00 | 1.00 | 1.00 |
| Multi-Krum | 1.00 | 1.00 | 1.00 | 1.00 |
| FoolsGold | 0.9972 | 1.00 | 1.00 | 1.00 |
| FLAME | 1.00 | 1.00 | 1.00 | 1.00 |
| CRFL | 0.9873 | 0.9892 | 0.9903 | 0.9884 |
| G-SPECTRE | 0.9948 | 1.00 | 1.00 | 1.00 |
| R-SPECTRE | 0.9899 | 0.9934 | 1.00 | 1.00 |
| Shadow Learning (label-level / user-level) | **0.0067** / 0.0216 | **0.0101** / 0.0367 | **0.0312** / 0.0769 | **0.0502** / 0.1338 |

is always above 0.940, and the MTA convergence rate of Algorithm 1 is always similar to the no defense scenario. We show the ASR for CIFAR-100 and Tiny-ImageNet in Appendix F.2.

Table 2: ASR for CIFAR-10 under continuous training shows the advantage of Algorithm 1, while others suffer backdoor leakage.

| Defense \ $\alpha$ | 0.15 | 0.25 | 0.35 | 0.45 |
|---|---|---|---|---|
| Noise Adding | 0.9212 | 0.9177 | 0.9388 | 0.9207 |
| Clipping and Noise Adding | 0.9142 | 0.9247 | 0.9282 | 0.9338 |
| RFA | 0.9353 | 0.9528 | 0.9563 | 0.9598 |
| Multi-Krum | 0.9254 | 0.9196 | 0.9219 | 0.9301 |
| FoolsGold | 0.8969 | 0.9038 | 0.9157 | 0.9143 |
| FLAME | 0.9328 | 0.9267 | 0.9291 | 0.9331 |
| CRFL | 0.8844 | 0.8731 | 0.8903 | 0.8891 |
| G-SPECTRE | 0.8754 | 0.9001 | 0.9142 | 0.9193 |
| R-SPECTRE | 0.7575 | 0.8826 | 0.8932 | 0.9091 |
| Shadow Learning | **0.0140** | **0.0355** | **0.0972** | **0.1865** |

## 5.2 Ablation study under distribution drift

We next provide an ablation study to illustrate why the components of shadow learning are all necessary. We run this evaluation under a distribution drift scenario, where the distribution changes every 400 rounds. We use homogeneous EMNIST in the first phase $e_1$, then reduce the number of samples with labels 2-5 to 10% of the original in phase $e_2$, i.e., we reduce the number of images with the label in $\{2, 3, 4, 5\}$ from 10 to 1 for each client. Again, the backdoor target label is 1.

We compare shadow learning with two simpler variants: Periodic R-SPECTRE and Shadow Network Prediction. In periodic R-SPECTRE, the backbone $N$ is retrained from scratch with SPECTRE every $R = 400$ rounds; filtering is done in every round. In Shadow Network Prediction, we use the same training as our framework, but prediction only uses the shadow network (as opposed to using the backbone network as in Algorithm 1). The MTA-ASR tradeoffs of all three algorithms in the second phase $e_2$ are shown in Fig. 3, where each marker represents a different point in time. Fig. 3 gives a more direct comparison of the convergence rates of ASR and MTA, which are critical to evaluating the efficiency of early-stopping. That is,

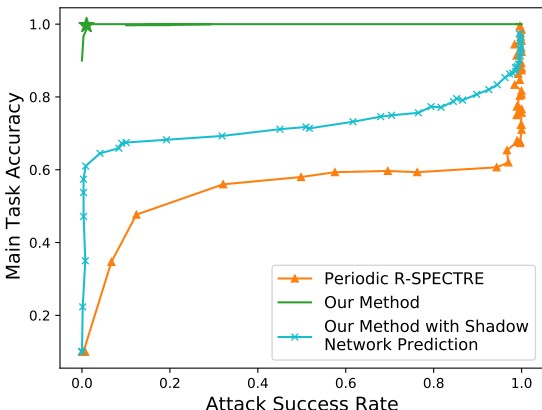

Figure 3: Ablation study for shadow learning under distribution drift. Simplified variants of our framework (e.g., periodic R-SPECTRE, shadow network prediction) significantly degrade the MTA-ASR Tradeoff with $\alpha = 0.15$, as achieving top-left is ideal.

early-stopping is effective if the curve reaches the top-left corner, i.e., high MTA and low ASR. Results for different periods and types of data distribution drift are shown in Appendix F.5.

Fig.3 shows that for $\alpha = 0.15$, full shadow learning achieves an MTA-ASR point of $(0.9972, 0.0103)$ (green star), but the simplified variants perform significantly worse. Shadow network prediction does better than periodic R-SPECTRE, as it conducts client filtering only after the convergence of backbone network $N$ on label $\ell$, while the latter filters in every round. This indicates that filtering after $N$ converges gives better performance. However, shadow network prediction suffers from the lack of training samples with labels 2-5, leading to poor performance.

## 5.3 Defense without knowing target label $\ell$

We next evaluate shadow learning when the defender only knows that the target label falls into a target set $S_\ell$ on EMNIST. In our generalized framework (Algorithm 5), we set the training round threshold as $R = 20$, and set the filtering ratio as $\kappa = 0.2$. To analyze the averaged ASR and target label detection success rate, we set $\alpha = 0.3$ and vary the size of $S_\ell$ from 2 to 10 (i.e., at 10, the defender knows nothing about $\ell$). For each experimental setting, we run our framework 20 times.

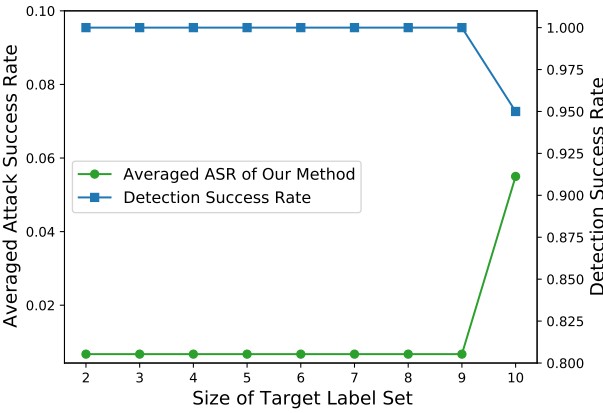

Figure 4: Defense Without Knowing the Exact Target Label with $\alpha = 0.3$

As observed in Figure 4, when $|S_\ell| \leq 9$, the averaged ASR of our method is near 0, and the target label can be detected successfully all the time, i.e., target label $\ell$ is in the filter target set $S'_\ell$. When $|S_\ell| = 10$, i.e.,

when the defender has no information about the target label, the averaged ASR and the detection success rates degrade gracefully (notice the different scales for the two vertical axes on Figure 4). This shows that **even without knowledge of the target label, shadow learning is still effective**.

To further demonstrate the performance of our algorithm on unknown target label defense, we set malicious rate as $\alpha = 0.15$ and conduct experiments on CIFAR-10, CIFAR-100 and Tiny-ImageNet datasets (with $|S_\ell| = 10, 100$ and 200 respectively). The attack success rate (ASR) and main task accuracy (MTA) are shown below.

Table 3: Unknown target label defense under different datasets with $\alpha = 0.15$

|  | CIFAR-10 | CIFAR-100 | Tiny-ImageNet |
|---|---|---|---|
| ASR | 0.084 | 0.099 | 0.092 |
| MTA | 0.948 | 0.862 | 0.801 |

Table 3 suggests that shadow learning still works on datasets with hundreds of labels while the defender has no information about the target label.

When the target label is unknown, shadow learning needs to train one shadow model for every label in the target set $S_\ell$, which could incur significant computational costs. However, these costs can be mitigated by the fact that each shadow model only needs to be trained for $R$ rounds ($R = 20$ in our experiments); after this, we find that we can determine the true target label with high accuracy and stop training the other shadow models. This assumes the target label is fixed, which may not be the case in practice.

To demonstrate the performance of our defense against multiple target label attacks (i.e., the attacker injects multiple backdoors targeting different labels in the model), we conduct experiments on Tiny-ImageNet, and vary the number of backdoors injected. We set the malicious rate to $\alpha = 0.15$ and assume the target label is totally unknown. The maximal attack success rate among all target labels (Max-ASR) and main task accuracy (MTA) is shown in Table 4.

Table 4: Defense against multiple target label attacks with $\alpha = 0.15$

| #Backdoor | Max-ASR | MTA |
|---|---|---|
| 1 | 0.092 | 0.801 |
| 50 | 0.107 | 0.794 |
| 100 | 0.125 | 0.813 |
| 150 | 0.122 | 0.783 |
| 200 | 0.150 | Not Applied |

From Table 4, we can observe that as the number of backdoors increases, Max-ASR increases slightly from 0.092 to 0.150 while MTA stays roughly the same, indicating that our algorithm can defend against attacks with multiple backdoors.

## 5.4 Defense under Client Heterogeneity

We finally evaluate the ASR of Algorithm 1 and its two variants, sample-level and user-level defenses, on the EMNIST dataset using the original dataset partition. In the sample level version, each user uploads the representations of samples with the target label without averaging, and the samples regarded as backdoor images are filtered out. This weakens privacy but is more robust against heterogeneous clients as shown in Table 5. On the other hand, user-level defense fails as heterogeneity of clients makes it challenging to detect corrupt users from user-level aggregate statistics. The label-level RFA and R-SPECTRE are evaluated after the network being trained for 1200 rounds. More experimental results with different heterogeneity levels and different datasets are provided in Appendix F.4.

Table 5: ASR under EMNIST partitioned as the original data

| Defense \ $\alpha$ | 0.15 | 0.25 | 0.35 | 0.45 |
|---|---|---|---|---|
| Label-level RFA | 1.00 | 1.00 | 1.00 | 1.00 |
| R-SPECTRE | 0.9967 | 1.00 | 1.00 | 1.00 |
| Shadow Learning (Sample-level) | **0.0107** | **0.0166** | **0.0301** | **0.0534** |
| Shadow Learning | 0.0307 | 0.0378 | 0.0836 | 0.1106 |
| Shadow Learning (User-level) | 0.5719 | 0.5912 | 0.6755 | 0.7359 |

### 5.5 Defense against attacks with explicit evasion of anomaly detection (Bagdasaryan et al., 2020)

We next evaluate shadow learning in the context of attacks that explicitly aim to evade anomaly detection techniques, namely Bagdasaryan et al. (2020). In this approach, to explicitly evade anomaly detection, adversarial clients modify their objective (loss) function by adding an anomaly detection term $\mathcal{L}_{ano}$:

$$\mathcal{L}_{model} = \gamma \mathcal{L}_{class} + (1 - \gamma)\mathcal{L}_{ano},$$

where $\mathcal{L}_{class}$ represents the accuracy on both the main and backdoor tasks. $\mathcal{L}_{ano}$ captures the type of anomaly detection they want to evade. In our setting, $\mathcal{L}_{ano}$ accounts for the difference between the representations of backdoor samples and benign samples with target label $\ell$. The ASR of our algorithm under this attack is shown in Table 6 with different values of $\gamma$. This experiment is run on the homogeneous EMNIST dataset.

Table 6: ASR of shadow learning under the Attack in Bagdasaryan et al. (2020)

| $\alpha$ | $\gamma = 0.4$ | $\gamma = 0.6$ | $\gamma = 0.8$ | $\gamma = 1$ |
|---|---|---|---|---|
| 0.15 | 0.0100 | 0.0100 | 0.0067 | 0.0067 |
| 0.25 | 0.0133 | 0.0167 | 0.0133 | 0.0101 |
| 0.35 | 0.0201 | 0.0304 | 0.0368 | 0.0312 |
| 0.45 | 0.0367 | 0.0702 | 0.0635 | 0.0502 |

We observe that with different $\gamma$, the ASR of shadow learning is always smaller than or equal to 0.07 for the homogeneous EMNIST dataset, indicating that the attacks cannot succeed under our defense method even with explicit evasion of anomaly detection.

### 5.6 Different backdoor trigger patterns

For completeness, we experiment with additional backdoor trigger patterns. We consider three different backdoor trigger patterns: diagonal trigger, random trigger, and periodic signal. The first two trigger patterns can be regarded as the pixel attack, while the last pattern belongs to the periodic attack (Barni et al., 2019). The diagonal trigger consists of black pixels in the top left to bottom right diagonal, and as for the random trigger, 25 pixels are randomly selected from the image and fixed as the trigger pattern. For the periodic signal, we choose the sine signal with amplitude 8 and frequency of 10. As shown in Table 7, under the homogeneous EMNIST dataset, the ASR of our method is smaller than 0.06 in all cases, indicating our method can generalize to different types of backdoor attacks.

### 5.7 Other settings

More experimental results including other datasets, defending against adaptive malicious rate attack strategies (F.6), robustness to differentially-private noise F.7, and sensitivity analysis of filtering algorithm hyperparameters F.8 are shown in Appendix F.

Table 7: ASR of Our Method under Different Trigger Patterns

| $\alpha$ | Diagonal Trigger | Random Trigger | Periodic Signal |
|---|---|---|---|
| 0.15 | 0.0133 | $0.00 \pm 0$ | 0.00 |
| 0.25 | 0.0234 | $0.0067 \pm 0.001$ | 0.0201 |
| 0.35 | 0.0281 | $0.0268 \pm 0.003$ | 0.0367 |
| 0.45 | 0.0569 | $0.0533 \pm 0.003$ | 0.0585 |

## 6  Conclusion

Motivated by the successes of filtering-based defenses against backdoor attack in the non-FL setting, we propose a novel and general framework for defending against backdoor attacks in FL under continuous training. Any FL-compatible filter can be plugged into our framework. The main idea is to use such filters to reduce the fraction of corrupt model updates significantly, and then train an early stopped model (called a shadow model) on those filtered updates to get a clean model. This combination of filtering and early-stopping significantly improves upon existing defenses and we provide a theoretical justification of our approach. One of the main technical innovations is the parallel training of the backbone and the shadow models, which is critical for obtaining a reliable filter (via the backbone model) and a trustworthy predictor (via the shadow model). Experimenting on four vision datasets and comparing against eight baselines, we show significant improvement on defense against backdoor attacks in FL.

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

# Appendix

## A    Related Work

For more comprehensive survey of backdoor attacks and defenses, we refer to Li et al. (2022b).

**Backdoor attacks.**    There has been a vast body of attacks on machine learning (ML) pipelines. In this work, we consider only *training-time* attacks, in which the attacker modifies either the training data or process to meet their goals, some of which are particularly deteriorating in FL settings (Sun et al., 2019; Wang et al., 2020b). Inference-time attacks are discussed in several survey papers (Madry et al., 2017; Ilyas et al., 2019). Further, we do not consider data poisoning attacks, in which the goal of the adversary is simply to decrease the model's prediction accuracy (Yang et al., 2017; Biggio et al., 2014) which has also been studied in the federated setting (Sun et al., 2019). Instead, we focus on *backdoor attacks*, where the attacker's goal is to train a model to output a target classification label on samples that contain a trigger signal (specified by the attacker) while classifying other samples correctly at test-time (Gu et al., 2017). The most commonly-studied backdoor attack is a *pixel attack*, in which the attacker inserts a small pattern of pixels into a subset of training samples of a given source class, and changes their labels to the target label (Gu et al., 2017). Pixel attacks can be effective even when only a small fraction of training data is corrupted (Gu et al., 2017). Many subsequent works have explored other types of backdoor attacks, including (but not limited to) periodic signals (Zhong et al., 2020), feature-space perturbations (Chen et al., 2017; Liu et al., 2017), reflections (Liu et al., 2020), strong trigggers that only require few-shot backdoor examples (Hayase & Oh, 2022), and FL model replacement that explicitly try to evade anomaly detection (Bagdasaryan et al., 2020). One productive line of work in in making the trigger human-imperceptible (Li et al., 2020b; Barni et al., 2019; Yao et al., 2019; Liu et al., 2017; Zhong et al., 2022; Feng et al., 2022; Wang et al., 2022a;d; Phan et al., 2022; Zhao et al., 2022a). Another line of work attempts to make the trigger stealthy in the latent space (Shokri et al., 2020; Cheng et al., 2021; Zhao et al., 2022b; Zeng et al., 2022; Qi et al., 2022; 2023). However, due to the privacy preserving nature of the federated setting, the attacker is limited in the information on other clients' data in the federated scenario. Although our evaluation will focus on pixel attacks, our experiments suggest that our insights translate to other types of attacks as well (Appendix 5.6 and 5.5). Advances in stronger backdoor attacks led to their use in other related domains, including copyright protection for training data (Li et al., 2020c; 2022c;a) and auditing differential privacy (Jagielski et al., 2020).

**Backdoor defenses.**    As mentioned earlier, many backdoor defenses can be viewed as taking one (or more) of three approaches: (1) Malicious data detection, (2) Robust training, and (3) Trigger identification.

Malicious data detection-based methods exploit the idea that adversarial samples will differ from the benign data distribution (e.g., they may be outliers). Many such defenses require access to clean samples (Liang et al., 2017; Lee et al., 2018; Steinhardt et al., 2017), which we assume to be unavailable. Others work only when the initial adversarial fraction is large, as in anti-backdoor learning (Li et al., 2021a) (see § 3), or small, as in robust vertical FL (Liu et al., 2021); we instead require a method that works across all adversarial fractions. Recently, *SPECTRE* proposed using robust covariance estimation to estimate the covariance of the benign data from a (partially-corrupted) dataset (Hayase et al., 2021). The data samples are whitened to amplify the spectral signature of corrupted data. Malicious samples are then filtered out by thresholding based on a *QUantum Entropy (QUE)* score, which projects the whitened data down to a scalar. We use SPECTRE as a building block of our algorithm, and also as a baseline for comparison. To adapt SPECTRE to the FL setting, we apply it to gradients (we call this G-SPECTRE) or sample representations (R-SPECTRE), each averaged over a single client's local data with target label $\ell$. However, we see in § 3 that SPECTRE alone does not work in the continuous learning setting. We propose shadow learning, where a second shadow model is trained on data filtered using one of the malicious data detection-based methods, for example Tran et al. (2018); Chen et al. (2018); Hayase et al. (2021). Some other malicious data filtering approaches, such as Huang et al. (2019); Do et al. (2022); Chen et al. (2022b), require inspecting clients' individual training samples, which is typically not feasible in FL systems. Instead, filtering approaches, such as Ma et al. (2022); Tang et al. (2021); Tran et al. (2018); Chen et al. (2018), that are based on the *statistics* of the examples can potentially be used within our shadow model framework.

Robust training methods do not explicitly identify and/or filter outliers; instead, they modify the training (and/or testing) procedure to implicitly remove their contribution. For example, *Robust Federated Aggregation (RFA)* provides a robust secure aggregation oracle based on the geometric median (Pillutla et al., 2019). It is shown to be robust against data poisoning attacks both theoretically and empirically. However, it was not evaluated on backdoor attacks. In this work, we adopt RFA as a baseline, and show that it too suffers from backdoor leakage (§ 3). Federated Learning Provable defense framework (FLIP) provides a trigger reverse engineering based defense against backdoor attack under federated learning setting (Zhang et al., 2022). However, it was not evaluated on continuous training setting (attack happens at least 1,000 rounds for example). Other variants of robust training methods require a known bound on the magnitude of the adversarial perturbation; for example, randomized smoothing (Wang et al., 2020a; Weber et al., 2020) ensures that the classifier outputs the same label for all points within a ball centered at a particular sample. We assume the radius of adversarial perturbations to be unknown at training time. Other approaches again require access to clean data, which we assume to be unavailable. Examples include fine-pruning (Liu et al., 2018), which trains a pruned, fine-tuned model from clean data.

Trigger identification approaches typically examine the training data to infer the structure of a trigger (Chen et al., 2019; 2022a; Tao et al., 2022; Guo et al., 2021; Hu et al., 2021; Wang et al., 2022c; Xiang et al., 2020; Wang et al., 2022b; Chai & Chen, 2022; Harikumar et al., 2022; Yue et al., 2022; Guan et al., 2022). For example, NeuralCleanse (Wang et al., 2019) searches for data perturbations that change the classification of a sample to a target class. SentiNet (Chou et al., 2018) uses techniques from model interpretability to identify contiguous, salient regions of input images. These approaches are ill-suited to the FL setting, as they require fine-grained access to training samples and are often tailored to a specific type of backdoor trigger (e.g., pixel attacks).

Finally, note that there is a large body of work defending against data poisoning attacks (Sun et al., 2019; Pillutla et al., 2019; Blanchard et al., 2017a; Awan et al., 2021; Laskov, 2014; Tolpegin et al., 2020). In general, such defenses may not work against backdoor attacks. For example, we show in § 3 and § 5 that defenses against data poisoning attacks such as RFA (Pillutla et al., 2019), norm clipping (Sun et al., 2019), and noise addition (Sun et al., 2019) are ineffective against backdoor attacks, particularly in the continuous training setting.

There are other defenses include that are less explored including test-time backdoor defense (Guo et al., 2023) and model unlearning (Zeng et al., 2021).

**Continuous training.** Distribution shift and temporal data heterogeneity exist widely in real-word applications, e.g., traffic congestion monitoring (Xu & Mao, 2020) and healthcare monitoring (Brophy et al., 2021). In order to adapt to changing data distributions, models should be trained continuously, both in the federated and central settings (Hofer & Krempl, 2013). There are several works (Xu & Mao, 2020; Yu et al., 2022) focused on the issues raised by continuous training in FL, e.g., communication cost and algorithm design. In some applications, enterprises may want to update their model to perform well on the current distribution, as well as on previously-seen distributions (i.e., they want to prevent catastrophic forgetting). However, in this paper, we consider a simpler setting in which the central party only wants to perform well on the currently-seen distribution. Even in this simpler setting, we show that existing approaches are unable to resist backdoor attacks. Hence, we view solving the current problem as a precursor to fully solving the backdoored continual learning problem in the federated setting.

## B  Algorithm Details

In the client filtering process, the collected representations are projected down to a *k*-dimensional space by $\boldsymbol{U}$, then whitened to get $\tilde{\boldsymbol{h}}_j = \hat{\boldsymbol{\Sigma}}^{-1/2}(\boldsymbol{U}^T\boldsymbol{h}_j - \hat{\boldsymbol{\mu}}) \in \mathbb{R}^k$ for all $j \in \mathcal{C}_r$. The projection onto $\boldsymbol{U}$ is to reduce computational complexity, which is less critical for the performance of the filter. The whitening with clean covariance $\hat{\boldsymbol{\Sigma}}$ and clean mean $\hat{\boldsymbol{\mu}}$ ensures that the poisoned representations stand out from the clean ones and is critical for the performance of the filter. Based on the whitened representations, the server calculates QUE scores (which roughly translates as the scaled-norm of the whitened representation) for all clients and keeps

the clients with scores less than the threshold $T$ as $\mathcal{C}'_r$ (FILTER in line 15). The details of GETTHRESHOLD and the client filtering process, FILTER, are shown in Algorithms 2 and 3.

---

**Algorithm 2:** GETTHRESHOLD (SPECTRE-based instantiation)

**input** : representation $S = \{\boldsymbol{h}_i\}_{i=1}^n$, malicious rate upper bound $\bar{\alpha}$, dimension $k$, QUE parameter $\beta$.

$\boldsymbol{\mu}(S) \leftarrow \frac{1}{n}\sum_{i=1}^n \boldsymbol{h}_i$;
$S_1 = \{\boldsymbol{h}_i - \boldsymbol{\mu}(S)\}_{i=1}^n$;
$\boldsymbol{V}, \boldsymbol{\Lambda}, \boldsymbol{U} = \text{SVD}_k(S_1)$;
$S_2 \leftarrow \{\boldsymbol{U}^\top \boldsymbol{h}_i\}_{\boldsymbol{h}_i \in S}$;
$\hat{\boldsymbol{\Sigma}}, \hat{\boldsymbol{\mu}} \leftarrow \text{ROBUSTEST } (S_2, \bar{\alpha})$; (Diakonikolas et al., 2019)
$S_3 \leftarrow \left\{ \hat{\boldsymbol{\Sigma}}^{-1/2} \left(\bar{\boldsymbol{h}}_i - \hat{\boldsymbol{\mu}}\right) \right\}_{\bar{\boldsymbol{h}}_i \in S_2}$;
$\{t_i\}_{i=1}^n \leftarrow \text{QUESCORE}(S_3, \beta)$                     [Algorithm 4]
$T \leftarrow$ the $1.5\bar{\alpha}n$-th largest value in $\{t_i\}_{i=1}^n$;
**return** $\hat{\boldsymbol{\Sigma}}, \hat{\boldsymbol{\mu}}, T, \boldsymbol{U}$

---

**Algorithm 3:** FILTER (SPECTRE-based instantiation)

**input** : $S = \left\{\bar{\boldsymbol{h}}_i \in \mathbb{R}^k\right\}_{i=1}^n$, estimated covariance $\hat{\boldsymbol{\Sigma}}$, estimated mean $\hat{\boldsymbol{\mu}}$, threshold $T$, QUE parameter $\beta$.

$S' \leftarrow \left\{ \hat{\boldsymbol{\Sigma}}^{-1/2} \left(\bar{\boldsymbol{h}}_i - \hat{\boldsymbol{\mu}}\right) \right\}_{\bar{\boldsymbol{h}}_i \in S}$;
$\{t_i\} \leftarrow \text{QUESCORE}(S', \beta)$                     [Algorithm 4]
**return** clients with QUE-scores smaller than $T$

---

The details of the QUESCORE (Hayase et al., 2021) is shown in Algorithm 4

---

**Algorithm 4:** QUESCORE (Hayase et al., 2021)

**input** : $S = \left\{\tilde{\boldsymbol{h}}_i \in \mathbb{R}^k\right\}_{i=1}^n$, QUE parameter $\beta$.

$$t_i \leftarrow \frac{\tilde{\boldsymbol{h}}_i^\top Q_\beta \tilde{\boldsymbol{h}}_i}{Tr(Q_\beta)}, \quad \forall i \in [n],$$

where $Q_\beta = \exp\left( \frac{\beta(\widetilde{\Sigma} - \mathbf{I})}{\|\widetilde{\Sigma}\|_2 - 1} \right)$ and $\widetilde{\Sigma} = \frac{1}{n}\Sigma_{i=1}^n \tilde{\boldsymbol{h}}_i \tilde{\boldsymbol{h}}_i^\top$.
**return** $\{t_i\}_{i=1}^n$

---

## C   Shadow Learning Framework Without Knowledge of the Target Label

For simplicity, we presented Algorithm 1 using knowledge of the target label $\ell$. However, this knowledge is not necessary. In practice, an adversary may know that the target label falls into a target set $S_\ell \subseteq [L]$. Note that the defender has no information about the target class when $S_\ell = [L]$, i.e., when it contains all labels. Under this assumption, shadow learning generalizes to Algorithm 5.

At training time, the central server maintains a *backbone model* and multiple *shadow models*, each shadow model corresponding to a potential target label in $S_\ell$. The backbone model $N$ is trained continuously based on Algorithm 1. For each shadow model $N'_y$, where $y \in S_\ell$, it is first trained for $R$ rounds according to Algorithm 1, where each client uploads the averaged representation of samples with label $y$. To distinguish the actual target label, for every shadow network training round $r$, each shadow network $N'_y$ calculates the QUE-ratio $\gamma_y^{(r)} = \frac{Q_1}{Q_2}$, where $Q_1$(resp. $Q_2$) is the QUE-score averaged over clients with scores larger (resp.

smaller) than $T$, which is the QUE threshold calculated in Algorithm 1. Compared with non-target labels, the QUE-scores of malicious clients calculated based on the target label $\ell$ are much larger than that of benign clients, and therefore, $\gamma_\ell^{(r)}$ is larger than $\gamma_y^{(r)}$ ($y \in S_\ell \setminus \{\ell\}$) in most cases. After the shadow networks have been trained for $R$ rounds, the averaged QUE-ratio $\overline{\gamma}_y$ will be calculated for every $N_y'$, according to which the filtered target set $S_\ell'$ can be obtained:

$$S_\ell' \leftarrow \left\{ y | \overline{\gamma}_y \text{ is among the top } \lceil \kappa |S_\ell| \rceil \text{ largest values of } \{\overline{\gamma}_z\}_{z \in S_\ell} \right\},$$

where the hyper-parameter $\kappa$ is the filtering ratio. Then only the shadow networks with labels in the filter target set $S_\ell'$ will be trained according to Algorithm 1.

At test time, all unlabeled samples are first passed through backbone $N$. If the label prediction $y$ falls into the filtered target set $S_\ell'$, the sample is passed through the early-stopped shadow network $N_y'$, whose prediction is taken as the final output. We show that Algorithm 5 works well for most $S_\ell$ and degrades gracefully as the set increases in Figure 4.

---

**Algorithm 5:** Shadow learning framework (training) without knowing the exact target label

---

    **input** : target set $S_\ell$, training round threshold $R$, filtering ratio $\kappa$.

**1** Initialize the networks $N$, $\left\{ N_y' \right\}_{y \in S_\ell}$;

**2** $\Gamma_y \leftarrow 0, \ \forall y \in S_\ell$;

**3** Train the backbone network $N$ according to Algorithm 1;

**4** **for** *each shadow network training round $r$* **do**

**5**     **if** $r \leq R$ **then**

**6**         **for** *each shadow network $N_y'$ where $y \in S_\ell$* **do**

**7**             Obtain QUE-score for each client and threshold $T$, and train $N_y'$ according to Algorithm 1, where each client uploads the averaged representation of samples with label $y$;

**8**             $Q_1 \leftarrow$ QUE-score averaged over clients with scores larger than $T$;

**9**             $Q_2 \leftarrow$ QUE-score averaged over clients with scores smaller than $T$;

**10**             $\gamma_y^{(r)} = \frac{Q_1}{Q_2}$;

**11**             $\Gamma_y \leftarrow \Gamma_y + \gamma_y^{(r)}$;

**12**     **if** $r = R$ **then**

**13**         $\overline{\gamma}_y \leftarrow \frac{\Gamma_y}{R}, \ \forall y \in S_\ell$;

**14**         $S_\ell' \leftarrow \left\{ y | \overline{\gamma}_y \text{ is among the top } \lceil \kappa |S_\ell| \rceil \text{ largest values of } \{\overline{\gamma}_z\}_{z \in S_\ell} \right\}$;

**15**     **if** $r > R$ **then**

**16**         Train $N_y'$, where $y \in S_\ell'$, according to Algorithm 1;

---

# D   Complete Proofs of the Main Theoretical Results

We provide proofs of main results and accompanying technical lemmas. We use $c, c', C, C', \dots$ to denote generic numerical constants that might differ from line to line.

## D.1   Proof of Theorem 1

The proof proceeds in two steps. First, we show that under Assumption 1, the direction of the top eigenvector of the empirical covariance is aligned with the direction of the center of the poisoned representations (Lemma D.1). We next show that filtering with the quantum score significantly reduces the number of poisoned clients (Lemma D.2).

We let $\hat{\Sigma}$ be the output of the robust covariance estimator in Algorithm 2. After whitening by $\hat{\Sigma}^{-1/2}$, we let $\tilde{\Sigma}_c = \hat{\Sigma}^{-1/2} \Sigma_c \hat{\Sigma}^{-1/2}$, $\tilde{\Sigma}_p = \hat{\Sigma}^{-1/2} \Sigma_p \hat{\Sigma}^{-1/2}$, $\tilde{\Delta} = \hat{\Sigma}^{-1/2} \Delta$, $\tilde{\mu}_p = \hat{\Sigma}^{-1/2} \mu_p$, and $\tilde{\mu}_c = \hat{\Sigma}^{-1/2} \mu_c$.

The next lemma shows that as we have a larger separation $\|\tilde{\Delta}\|$, we get a better estimate of the direction of the poisons, $\tilde{\Delta}/\|\tilde{\Delta}\|$, using the principal component, $v$, of the whitened representations $S' = \{\Sigma^{-1/2}(h_i - \mu)\}_{i=1}^{n_r}$. The estimation error is measured in $\sin^2$ of the angle between the two.

**Lemma D.1** (Estimating $\tilde{\Delta}$ with top eigenvector). *Under the assumptions of Theorem 1,*

$$\sin^2(v, \tilde{\Delta}/\|\tilde{\Delta}\|) \;=\; 1 - (v^T \tilde{\Delta}/\|\tilde{\Delta}\|)^2 \;\leq\; c\frac{(\log(1/\alpha) + \xi)^2}{\|\tilde{\Delta}\|^4} \;. \tag{1}$$

We next show that when projected onto any direction $v$, the number of corrupted client updates passing the filter is determined by how closely aligned $v$ is with the direction of the poisoned data, $\tilde{\Delta}$, and the magnitude of the separation, $\|\tilde{\Delta}\|$.

**Lemma D.2** (Quantum score filtering). *Under the hypotheses of Theorem 1,*

$$\frac{|S_{\text{poison}} \setminus S_{\text{filter}}|}{|(S_{\text{poison}} \cup S_{\text{clean}}) \setminus S_{\text{filter}}|} \;\leq\; c' \alpha \, n_r \, Q\Big(\frac{v^T \tilde{\Delta} - c\sqrt{\log(1/\alpha)}}{\xi^{1/2}}\Big) \;, \tag{2}$$

*where $Q(t) = \int_t^\infty (1/\sqrt{2\pi})e^{-\frac{x^2}{2}} dx$ is the tail of the standard Gaussian.*

Since $\|\tilde{\Delta}\|^2 \geq C(\log(1/\alpha) + \xi)$, Lemma D.1 implies $v^T \tilde{\Delta} \geq (1/2)\|\tilde{\Delta}\|$. Since $(v^T \tilde{\Delta} - c\sqrt{\log(1/\alpha)})/\xi^{1/2} \geq C'\sqrt{\log(1/\alpha)/\xi}$, Lemma D.2 implies that $|S_{\text{poison}} \setminus S_{\text{filter}}| \leq \alpha^{C''/\xi}$. Since $\xi < 1$, we can make any desired exponent by increasing the separation by a constant factor.

### D.1.1  Proof of Lemma D.1

Let $\tilde{\Sigma}$ denote the empirical covariance of the whitened representations. With a large enough sample size, we have the following bound on the robustly estimated covariance.

**Theorem 2** (Robust covariance estimation (Diakonikolas et al., 2017)[Theorem 3.3]). *If the sample size is $n_r = \Omega((d^2/\alpha^2)\text{polylog}(d/\alpha))$ with a large enough constant then with probability $9/10$,*

$$\|\hat{\Sigma}^{-1/2}\Sigma_c\hat{\Sigma}^{-1/2} - \mathbf{I}_d\|_F \;\leq\; c\alpha \log(1/\alpha) \;, \tag{3}$$

*for some universal constant $c > 0$ where $\|A\|_F$ denotes the Frobenius norm of a matrix $A$.*

Denoting $\tilde{\Sigma} = \hat{\Sigma}^{-1/2}\Sigma_{\text{emp}}\hat{\Sigma}^{-1/2}$ with the empirical covariance of the clean representations denoted by $\Sigma_{\text{emp}} = (1/n)\sum_{i=1}^n (h_i - \mu_{\text{emp}})(h_i - \mu_{\text{emp}})^T = (1-\alpha)\hat{\Sigma}_C + \alpha\hat{\Sigma}_p + \alpha(1-\alpha)\hat{\Delta}\hat{\Delta}^T$, where $\hat{\Sigma}_c$, $\hat{\Sigma}_p$, and $\hat{\Delta}$ are the empirical counterparts, we can use this to bound,

$$\begin{aligned}
&\|\tilde{\Sigma} - (1-\alpha)\mathbf{I}_d - \alpha\tilde{\Sigma}_p - \alpha(1-\alpha)\tilde{\Delta}\tilde{\Delta}^T\| \\
&\leq\; (1-\alpha)\|\hat{\Sigma}^{-1/2}(\hat{\Sigma}_c - \Sigma_c)\hat{\Sigma}^{-1/2}\| + (1-\alpha)\|\hat{\Sigma}^{-1/2}\Sigma_c\hat{\Sigma}^{-1/2} - \mathbf{I}_d\| \\
&\quad +\; \alpha\|\hat{\Sigma}^{-1/2}(\Sigma_p - \hat{\Sigma}_p)\hat{\Sigma}^{-1/2}\| + \alpha(1-\alpha)\|\hat{\Sigma}^{-1/2}(\Delta\Delta^T - \hat{\Delta}\hat{\Delta}^T)\hat{\Sigma}^{-1/2}\| \\
&\leq\; c'\alpha \log(1/\alpha) \;,
\end{aligned} \tag{4}$$

for a large enough sample size $n_r = \tilde{\Omega}(d^2/\alpha^3)$. Among other things, this implies that

$$\|\tilde{\Sigma} - (1-\alpha)\mathbf{I}\| \;\geq\; \alpha(1-\alpha)\|\tilde{\Delta}\|^2 - c'\alpha \log(1/\alpha) \;. \tag{5}$$

We use Davis-Kahan theorem to turn a spectral norm bound between two matrices into a angular distance bound between the top singular vectors of the two matrices:

$$
\begin{aligned}
\sqrt{1 - \frac{(v^T\tilde{\Delta})^2}{\|\tilde{\Delta}\|^2}} \;&\leq\; \frac{\|(\|\tilde{\Sigma}\| - (1-\alpha))vv^T - \alpha(1-\alpha)\tilde{\Delta}\tilde{\Delta}^T\|_F}{\|\tilde{\Sigma} - (1-\alpha)\mathbf{I}\|} \\[2mm]
&\leq\; \frac{\sqrt{2}\|(\|\tilde{\Sigma}\| - (1-\alpha))vv^T - \alpha(1-\alpha)\tilde{\Delta}\tilde{\Delta}^T\|}{\|\tilde{\Sigma} - (1-\alpha)\mathbf{I}\|} \\[2mm]
&\leq\; \frac{2\sqrt{2}\|\tilde{\Sigma} - (1-\alpha)\mathbf{I} - \alpha(1-\alpha)\tilde{\Delta}\tilde{\Delta}^T\|}{\|\tilde{\Sigma} - (1-\alpha)\mathbf{I}\|} \\[2mm]
&\leq\; \frac{c\alpha(\log(1/\alpha) + \|\tilde{\Sigma}_p\|)}{\alpha(1-\alpha)\|\tilde{\Delta}\|^2 - c'\alpha\log(1/\alpha)} \;,
\end{aligned}
\tag{6}
$$

where $\|A\|$ and $\|A\|_F$ denote spectral and Frobenius norms of a matrix $A$, respectively, the first inequality follows from Davis-Kahan theorem (Davis & Kahan, 1970), the second inequality follows from the fact that Frobenius norm of a rank-2 matrix is bounded by $\sqrt{2}$ times the spectral norm, and the third inequality follows from the fact that $(\|\tilde{\Sigma}\| - (1-\alpha))vv^T$ is the best rank one approximation of $\tilde{\Sigma} - (1-\alpha)\mathbf{I}$ and hence $\|(\|\tilde{\Sigma}\| - (1-\alpha))vv^T - \tilde{\Delta}\tilde{\Delta}^T\| \leq \|(\|\tilde{\Sigma}\| - (1-\alpha))vv^T - (\tilde{\Sigma} - (1-\alpha)\mathbf{I})\| + \|(\tilde{\Sigma} - (1-\alpha)\mathbf{I}) - \tilde{\Delta}\tilde{\Delta}^T\| \leq 2\|\tilde{\Sigma} - (1-\alpha)\mathbf{I} - \tilde{\Delta}\tilde{\Delta}^T\|$. The last inequality follows from Eq. (4) and Eq. (5). For $\alpha \leq 1/2$ and $\|\tilde{\Delta}\| \geq \sqrt{\log(1/\alpha)}$, this implies the desired result.

### D.1.2 Proof of Lemma D.2

We consider a scenario where the QUEscore of a (whitened and centered) representation $\tilde{h}_i = \hat{\Sigma}^{-1/2}(h_i - \hat{\mu}_c)$, where $\hat{\mu}_c$ is the robust estimate of $\mu_c$, is computed as

$$
\tau_i^{(\beta)} \;=\; \frac{\tilde{h}_i^T Q_\beta \tilde{h}_i}{\text{Tr}(Q_\beta)} \;,
\tag{7}
$$

where $Q_\beta = \exp((\alpha/\|\tilde{\Sigma}\| - 1)(\tilde{\Sigma} - \mathbf{I}))$. We analyze the case where we choose $\beta = \infty$, such that $\tau_i^{(\infty)} = (v^T\tilde{h}_i)^2$ and the threshold $T$ returned by SPECTRE satisfies the following. If we have infinite samples and there is no error in the estimates $v$, $\tilde{\Sigma}$, and $\hat{\mu}_c$, then we have $Q^{-1}((3/4)\alpha) \leq T^{1/2} \leq Q^{-1}((1/4)\alpha)$, which follows from the fact that for the clean data with identity covariance, we can filter out at most $1.5\alpha$ fraction of the data (which happens if we do not filter out any of the poisoned data points) and we can filter out at least $0.5\alpha$ fraction of the data (which happens if we filter out all the poisoned data). With finite samples and estimation errors in the robust estimates, we get the following:

$$
Q^{-1}((3/4)\alpha + \alpha^2/d) - c'\alpha/d \;\leq\; T^{1/2} \;\leq\; Q^{-1}((1/4)\alpha - c'\alpha^2/d) + c'\alpha/d \;,
\tag{8}
$$

where $Q(\cdot)$ is the tail of a standard Gaussian as defined in Lemma D.2 and we used the fact that for a large enough sample size we have $\|v^T\hat{\Sigma}^{-1/2}(\mu_c - \hat{\mu}_c)\| \leq c'\alpha/d$.

At test time, when we filter out data points with QUEscore larger than $T$, we have that we filter out at most clean $|S_{\text{clean}} \cap S_{\text{filter}}| \leq 2\alpha n_r$ representations for a large enough $d$. Similarly, we are guaranteed that the remaining poisoned representations are at most $|S_{\text{poison}} \setminus S_{\text{filter}}| \leq Q((v^T\tilde{\Delta} - T^{1/2})/\xi^{1/2})(\alpha n_r)$. Since from above bound $T^{1/2} \leq c\sqrt{\log(1/\alpha)}$, this proves the desired bound.

## D.2 Assumptions for Corollary 4.1

Corollary 4.1 follows as a corollary of Li et al. (2020a)[Theorem 2.2]. This critically relies on an assumption on a $(\varepsilon_0, M)$-clusterable dataset $\{(x_i \in \mathbb{R}^{d_0}, y_i \in \mathbb{R})\}_{i=1}^{n_r}$ and overparametrized two-layer neural network models, as defined in Assumption 2 below.

**Assumption 2** $((\alpha, n_r, \varepsilon_0, \varepsilon_1, M, L, \hat{M}, C, \lambda, W)$-model)**.** *The $(1-\alpha)$ fraction of data points are clean and originate from $M$ clusters with each cluster containing $n_r/M$ data points. Cluster centers are unit norm vectors, $\{\mu_q \in \mathbb{R}^{d_0}\}_{q=1}^M$. An input $x_i$ that belong to the $q$-th cluster obeys $\|\boldsymbol{x}_i - \mu_q\| \leq \varepsilon_0$, with $\varepsilon_0$ denoting*

*the input noise level. The labels $y$ belong to one of the $L$ classes and we place them evenly in $[-1, 1]$ in the training such that labels correspond to $y \in \{-1, -1 + 1/(L-1), \ldots, 1\}$. We let $C = [\mu_1, \ldots, \mu_M]^T \in \mathbb{R}^{M \times d_0}$ and define $\lambda = \lambda(C)$ as the minimum eigenvalue of the matrix $CC^T \odot \mathbb{E}[\phi'(Cg)\phi'(Cg)^T]$ for $g \sim \mathcal{N}(0, \mathbf{I}_{d_0})$.*

*All clean data points in the same cluster share the same label. Any two clusters obey $|\mu_q - \mu_{q'}| \geq 2\varepsilon_0 + \varepsilon_1$, where $\varepsilon_1$ is the size of the trigger. The corrupted data points are generated from a data point $\boldsymbol{x}_i$ with a source label $y_i = y_{\text{source}}$ belonging to one of the clusters for the source. A fixed trigger $\boldsymbol{\delta}$ is added to the input and labelled as a target label $q$ such that the corrupted paired example is $(\mathbf{x}_i + \boldsymbol{\delta}, q)$. We train on the combined dataset with $\alpha n_r$ corrupted points and $(1 - \alpha)n_r$ uncorrupted points. We train a neural network of the form $f_W(\boldsymbol{x}) = v^T \phi(W\boldsymbol{x})$ for a trainable parameter $W \in \mathbb{R}^{\hat{M} \times d_0}$ and a fixed $v \in \mathbb{R}^{\hat{M}}$, where the head $v$ is fixed as $1/\sqrt{\hat{M}}$ for half of the entries and $-1/\sqrt{\hat{M}}$ for the other half. We assume the activation function $\phi : \mathbb{R} \to \mathbb{R}$ satisfy $|\phi'(z)|, |\phi''(z)| < c''$ for some constant $c'' > 0$.*

## D.3 Necessity of robust covariance estimation

To highlight that SPECTRE, and the robust covariance estimation, is critical in achieving this guarantee, we next show that under the same assumptions, using the QUEscore filtering without whitening fails and also using whitening with non-robust covariance estimation also fails, in the sense that the fraction of the corrupted data is non-decreasing. We construct an example within Assumption 1 as follows: $\mu_c = 0$ and $\Sigma_c = \sigma^2(\mathbf{I} - (1 - \delta)uu^T)$ for a unit norm $u$. We place all the poisons at $\mu_p = au$ with $\xi = 0$ and covariance $\Sigma_p$ zero. We let $\delta \leq a/(c_m^2 \log(1/\alpha))$ such that the separation condition is met. By increasing $\sigma^2$, we can make the inner product of top PCA direction $v_{\text{combined}}$ of the combined data and the direction of the poisons $u$ arbitrarily small. Hence, after finding the threshold $T$ in the projected representations $v_{\text{combined}}^T h_i$ and using this to filter out the representations, as proposed in Tran et al. (2018), the ratio of the poisons can only increase as all the poisons are placed close to the center of the clean representations after projection. The same construction and conclusion holds for the case when we whitened with not the robustly estimated covariance, but the QUEscore based filter with $\beta = \infty$ projects data first onto the PCA direction of the whitened data, which can be made arbitrarily orthogonal to the direction of the poisons, in high dimensions, i.e. $v_{\text{whitened}}^T u \leq 2/d$ with high probability. This follows from the fact that after (non-robust) whitening, all directions are equivalent and the chance of PCA finding the direction of the poisons is uniformly at random over all directions in the $d$ dimensional space.

# E Experimental Details

In the experiments in §5 and Appendix F, we train on the EMNIST, CIFAR-10, CIFAR-100, and Tiny-ImageNet datasets. In the EMNIST dataset, there are 3383 users with roughly 100 images in ten labels per user with heterogeneous label distributions. We train a convolutional network with two convolutional layers, max-pooling, dropout, and two dense layers. The CIFAR-10 dataset has $50,000$ training examples, with 5000 samples in each label. We partition those samples to 500 users uniformly at random and train a ResNet-18 (He et al., 2016). With 500 samples in each label, the CIFAR-100 and Tiny-ImageNet datasets have $50,000$ and $100,000$ training examples respectively. For both datasets, we partition those samples to 100 users uniformly at random and train a ResNet-18.

For all datasets, the server randomly selects 50 clients each round, and each client trains the current model with the local data with batch size 20, learning rate 0.1, and for two iterations. The server learning rate is 0.5. The attacker tries to make 7's predicted as 1's for EMNIST, horses as automobiles for CIFAR-10, roses as dolphin for CIFAR-100, and bees as cats for Tiny-ImageNet. The backdoor trigger is a $5 \times 5$-pixel black square at the bottom right corner of the image. An $\alpha$ fraction of the clients are chosen to be malicious, who are given 10 corrupted samples. We set the malicious rate $\alpha$ as its upper bound, i.e., $\alpha = \bar{\alpha}$. We study first homogeneous and static settings, and we discuss heterogeneous and dynamic settings in §5.4 and §5.2. In our framework, we set the retraining threshold $\epsilon_1$ as $2\%$, and the convergence threshold $\epsilon_2$ as $0.05\%$. We let the dimensionality reduction parameter $k$ be 32 and set the QUE parameter $\beta$ as 4 in Algorithm 1.

**Baselines.** SPECTRE (Hayase et al., 2021) adopts robust covariance estimation and data whitening to amplify the spectral signature of corrupted data, and then detects backdoor samples based on quantum

entropy (QUE) score. In federated learning settings, we adopt gradient- and representation-based SPECTRE, which takes as input the gradient updates or sample representations averaged over a single client's local data with target label $\ell$. For both versions, we conduct the robust estimation and quantum score filtering every training round regardless of the computation constraints. We let the dimensionality reduction parameter $k$ be 32 and set the QUE parameter $\beta$ as 4.

RFA (Pillutla et al., 2019) provides a robust secure aggregation oracle based on the geometric median, which is calculated by the Weiszfeld's Algorithm. In our experiments, we implement RFA with 4-iteration Weiszfeld's Algorithm. We also consider the label-level RFA: for each label, the geometric median of the aggregated gradient uploaded from each client is estimated. The server then updates the model based on the averaged geometric medians of all labels.

Norm Clipping defense (Sun et al., 2019) bounds the norm of each model update to at most some threshold $M$, and Noise Adding method (Sun et al., 2019) is to add a small amount of Gaussian noise to the updates. In our experiments, we set the norm threshold $M$ as 3 and add independent Gaussian noise with variance 0.03 to each coordinate.

Multi-Krum (Blanchard et al., 2017b) provides an aggregation rule that only select a subset of uploaded gradients that are close to most of their neighbors. In our experiments, we set the Byzantine parameter $f$ as $50\bar{\alpha}$, where $\bar{\alpha}$ is the malicious rate upper bound, and set the number of selected gradients as $m = 20$.

FoolsGold (Fung et al., 2018) provides an aggregation method that uses an adaptive learning rate per client based on inter-client contribution similarity. In our experiments, we set the confidence parameter as 1.

FLAME (Nguyen et al., 2022) adopts noise adding method, and uses the model clustering and norm clipping approach to reduce the amount of noise added. In our experiments, we set the noise level factors as $\lambda = 0.001$.

CRFL (Xie et al., 2021) trains a certifiably robust FL model by adopting noise adding and norm clipping during the training time and using randomized parameter smoothing during testing. In our experiments, we set the norm threshold $M$ as 3 and add independent Gaussian noise with variance 0.03 to each coordinate during training. We use $\sigma_T = 0.01$ to generate $M = 500$ noisy models in parameter smoothing procedure, and set the certified radius as 1 and the error tolerance as 0.001.

### E.0.1 Resource Costs

All algorithms including ours are implemented and performed on a server with two Xeon Processor E5-2680 CPUs. Running all defenses for our experiments took approximately 1000 CPU-core hours.

## F Additional Experimental Results

### F.1 Backdoor leakage under more datasets

Under the CIFAR-10, CIFAR-100, and Tiny-ImageNet datasets, we show that *backdoor leakage* phenomenon also results in the failure of existing backdoor defenses in Figure 5. Similar to Figure 1(a), we fix the malicious rate as $\alpha = 3\%$ and run the experiments for $12,000$ training rounds under the static setting where data distribution is fixed over time. We can observe that the attack success rate (ASR) of all competing defenses eventually approach around 1, 0.85, and 0.7 for each dataset, while the ASR of our algorithm keeps near 0 ($0.009, 0.027$, and $0.030$ for each dataset) all the time.

### F.2 Defense under homogeneous and static clients with more datasets

We evaluate our shadow learning framework under CIFAR-100 and Tiny-ImageNet datasets in Table 8 and 9. We can observe that all existing defenses suffer from backdoor leakage. CIFAR-100 and Tiny-ImageNet are also more difficult to learn compared with EMNIST dataset, and therefore the ASRs of our shadow learning framework in Table 8 and 9 are slightly larger than those in Table 1. Besides, with different $\alpha$, the MTA of Algorithm 1 is always above 0.85 and 0.70 for CIFAR-100 and Tiny-ImageNet respectively, and the MTA convergence rate of Algorithm 1 is always similar to the no defense scenario.

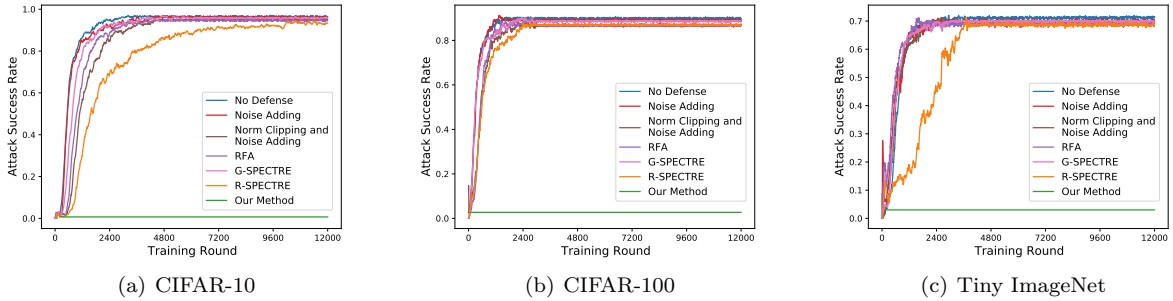

(a) CIFAR-10        (b) CIFAR-100        (c) Tiny ImageNet

Figure 5: Backdoor leakage causes competing defenses to fail eventually, even with a small $\alpha = 3\%$.

Table 8: ASR for CIFAR-100 under continuous training shows the advantage of Algorithm 1.

| Defense $\setminus$ $\alpha$ | 0.15 | 0.25 | 0.35 | 0.45 |
|---|---|---|---|---|
| Noise Adding | 0.8688 | 0.8634 | 0.8525 | 0.8743 |
| Clipping and Noise Adding | 0.8593 | 0.8604 | 0.8629 | 0.8688 |
| RFA | 0.8642 | 0.8697 | 0.8739 | 0.8697 |
| Multi-Krum | 0.8576 | 0.8594 | 0.8635 | 0.8741 |
| FoolsGold | 0.8107 | 0.8251 | 0.8299 | 0.8415 |
| FLAME | 0.8361 | 0.8592 | 0.8688 | 0.8691 |
| CRFL | 0.8142 | 0.8197 | 0.8033 | 0.8251 |
| G-SPECTRE | 0.8415 | 0.8597 | 0.8542 | 0.8673 |
| R-SPECTRE | 0.7978 | 0.8033 | 0.8306 | 0.8467 |
| Shadow Learning | **0.0765** | **0.0929** | **0.1694** | **0.2247** |

Table 9: ASR for Tiny-ImageNet under continuous training shows the advantage of Algorithm 1.

| Defense $\setminus$ $\alpha$ | 0.15 | 0.25 | 0.35 | 0.45 |
|---|---|---|---|---|
| Noise Adding | 0.7079 | 0.6920 | 0.7253 | 0.9139 |
| Clipping and Noise Adding | 0.6839 | 0.6907 | 0.6971 | 0.7182 |
| RFA | 0.7164 | 0.7206 | 0.7091 | 0.6981 |
| Multi-Krum | 0.6841 | 0.7193 | 0.7032 | 0.7167 |
| FoolsGold | 0.6693 | 0.6872 | 0.7013 | 0.6944 |
| FLAME | 0.6767 | 0.6931 | 0.6792 | 0.6784 |
| CRFL | 0.6360 | 0.6519 | 0.6440 | 0.6279 |
| G-SPECTRE | 0.6423 | 0.6691 | 0.6945 | 0.6932 |
| R-SPECTRE | 0.5519 | 0.6090 | 0.6495 | 0.6826 |
| Shadow Learning | **0.0720** | **0.0799** | **0.1127** | **0.1519** |

### F.3 Ablation study: Main Task Accuracy (MTA) vs. Attack Success Rate (ASR) Tradeoff

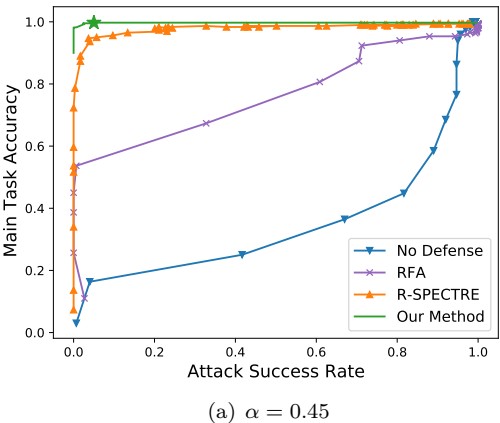
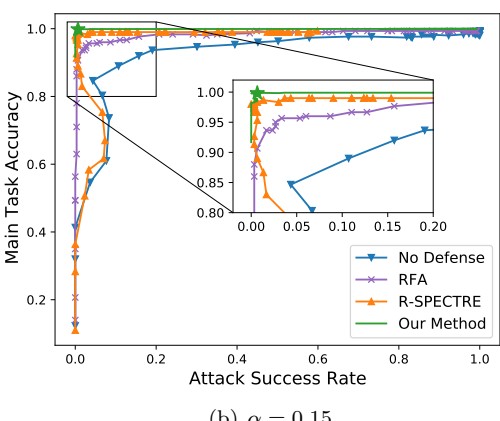

(a) $\alpha = 0.45$                               (b) $\alpha = 0.15$

Figure 6: MTA-ASR tradeoff shows simple early stopping with no (other) defense far from the ideal (0,1). R-SPECTRE with early stopping is more resilient, but all early-stopping-based prior defenses suffer when clients' data changes dynamically as we show in §5.2. Algorithm 1 achieves close-to-ideal tradeoffs.

Under homogeneous and static EMNIST dataset, we run an ablation study on one of the main components: the backbone network. Without the backbone network, our framework (in the one-shot setting) reduces to training baseline defenses with early stopping. Figure 6 shows the resulting achievable (ASR, MTA) as we tune the early-stopping round for $\alpha = 0.45$ and 0.15. The curves start at the bottom left (0,0) and most of them first move up, learning the main tasks. Then, the curves veer to the right as the backdoor is learned. We want algorithms that achieve points close to the top left $(0, 1)$.

When $\alpha = 0.45$, Algorithm 1 achieves the green star. The blue curve (early stopping with no other defense) is far from $(0, 1)$. This suggests that the backbone network and SPECTRE filtering are necessary to achieve the performance of Algorithm 1. The curve for early-stopped RFA (purple x's) is also far from $(0, 1)$ for large $\alpha$. Early-stopped R-SPECTRE without the backbone network (orange triangles) achieves a good MTA-ASR tradeoff (though still worse than that of Algorithm 1). However, we show in §5.2 that the MTA-ASR tradeoff of R-SPECTRE is significantly worse when clients' data distribution changes dynamically.

When $\alpha = 0.15$, the learning rate of backdoor samples is much smaller than the main task learning rate for all curves. However, the curves for early-stopping with no defense and early-stopped RFA still cannot achieve close-to-ideal tradeoffs.

### F.4 Synthetic heterogeneous clients

In synthetic heterogeneous EMNIST, each client receives shuffled images from 4 randomly selected classes with 25 samples per class. As shown in Table 10, it has a similar trend as the naturally heterogeneous dataset from the original EMNIST shown in Table 5.

Further, we analyze the situation where the user-level version of our algorithm works. We construct variants of EMNIST dataset with different heterogeneity level $h$. We call the dataset $h$-heterogeneous if the first $h$-fraction of the overall training images are shuffled and evenly partitioned to each client, and for the remaining $(1 - h)$-fraction samples, each client receives shuffled images from 4 randomly selected classes with $\lfloor 25(1 - h) \rfloor$ samples per class. For the adversarial clients, they also own 10 backdoor images. We fix the malicious rate as $\alpha = 0.15\%$. Table 11 shows the ASR of the user-level version of our algorithm under the dataset with different heterogeneity levels.

The ASR of the user-level version of our algorithm is smaller than 0.1 when $h \leq 0.4$, indicating that our user-level method can achieve the defense goal when under the dataset with low heterogeneity level. However,

Table 10: ASR under synthetic heterogeneous EMNIST

| $\alpha$ | Our Method with Sample-level Defense | Our Method | Our Method with User-level Defense |
|---|---|---|---|
| 0.15 | 0.0134 | 0.0334 | 0.6756 |
| 0.25 | 0.0268 | 0.0401 | 0.7424 |
| 0.35 | 0.0535 | 0.0970 | 0.8127 |
| 0.45 | 0.0669 | 0.1305 | 0.9833 |

when the heterogeneity level is high, the malicious clients cannot be distinguished by the aggregated statistic over all local samples, which results in the failure of the user-level defense method.

Table 11: ASR of Our Method in User-level Version with $\alpha = 0.15$

| Heterogeneity Level $h$ | 0 | 0.2 | 0.4 | 0.6 | 0.8 | 1 |
|---|---|---|---|---|---|---|
| ASR | 0.0216 | 0.0334 | 0.0969 | 0.2508 | 0.4816 | 0.6756 |

We also analyze the ASR and MTA of our algorithm (label-level version) under the datasets CIFAR-100 and Tiny-ImageNet with different heterogeneity level $h$. We fix the malicious rate as $\alpha = 0.15$. As shown in Table 12 and 13, the ASR is always smaller than 0.11 under any heterogeneity level for both datasets, while the MTA is always larger than 0.85 and 0.76 for CIFAR-100 and Tiny-ImageNet respectively.

Table 12: ASR and MTA of Our Method under synthetic heterogeneous CIFAR-100 with $\alpha = 0.15$

| Heterogeneity Level $h$ | 0 | 0.2 | 0.4 | 0.6 | 0.8 | 1 |
|---|---|---|---|---|---|---|
| ASR | 0.077 | 0.079 | 0.086 | 0.091 | 0.094 | 0.103 |
| MTA | 0.864 | 0.859 | 0.861 | 0.858 | 0.851 | 0.853 |

### F.5   More experiments under the dynamic setting

Recall the initial phases $e_1$ and $e_2$ in Section 5.2. In the third phase $e_3$, for all clients, we reduce the number of samples with labels 0 and 1 to 10% of the original while keeping other samples at the same level in $e_1$, i.e., each client has one image for label 0 and 1 respectively, and ten images for any other label. This tradeoff is shown in Figure 7. Since the number of samples with the target label $\ell = 1$ is reduced, the learning rates of both the backdoor samples and the benign samples with the target label decrease. This is similar to the setting in Figure 6(b) except that the proportion of samples with the target label is smaller.

We also consider the case where the data distribution over malicious clients is fixed over time, which rarely happens in practice. In a new phase $e_4$, we let each benign client contain 5 images for label 0 and 1 respectively, and 10 images for any other label. In phase $e_5$, we further reduce the number of images each benign client contains for label 0 or 1 to one. However, the local dataset of each adversarial client is always fixed as the malicious dataset in $e_1$, i.e., 10 backdoor images in target label $\ell = 1$ and 90 images for other labels. The MTA-ASR tradeoffs of phases $e_4$ and $e_5$ are shown in Fig.8

Our algorithm detects the data distribution change and renews the retraining period at the beginning of both phases. The MTA-ASR point our algorithm achieves is $(0.9944, 0.1471)$ and $(0.9937, 0.7458)$ in phases $e_4$ and $e_5$ respectively. The MTA-ASR tradeoffs of the comparing algorithms are much worse than ours in both phases.

Table 13: ASR and MTA of Our Method under synthetic heterogeneous Tiny-ImageNet with $\alpha = 0.15$

| Heterogeneity Level $h$ | 0 | 0.2 | 0.4 | 0.6 | 0.8 | 1 |
|---|---|---|---|---|---|---|
| ASR | 0.072 | 0.071 | 0.078 | 0.083 | 0.089 | 0.091 |
| MTA | 0.793 | 0.767 | 0.798 | 0.802 | 0.789 | 0.794 |

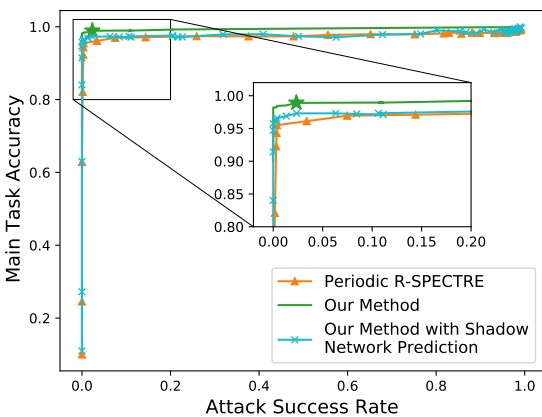

Figure 7: Analysis of MTA-ASR Tradeoff with $\alpha = 0.15$ after distribution shift in phase $e3$.

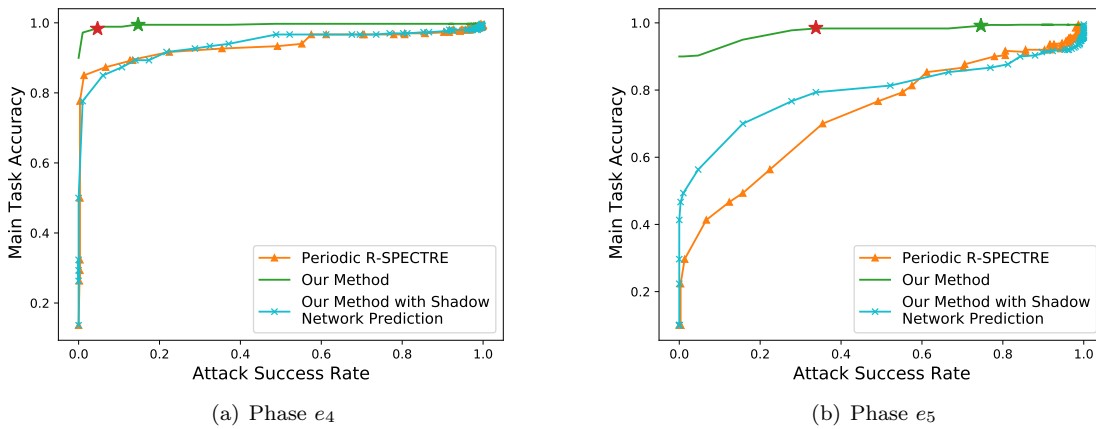

(a) Phase $e_4$

(b) Phase $e_5$

Figure 8: Analysis of MTA-ASR Tradeoff with $\alpha = 0.15$ under Time-varying Dataset and Static Adversarial Clients

Since the benign features with label 1 becomes more difficult to learn due to the leak of samples, and the proportion of backdoor samples among samples with target labels increases to 26% and 64% in phases $e_4$ and $e_5$ respectively, it is difficult to achieve the close-to-ideal MTA-ASR tradeoff. If we increase the convergence threshold $\epsilon_2$ from 0.05% to 0.5% when determining the early-stopping point, we can obtain the new MTA-ASR point indicated by the red star as $(0.9834, 0.0418)$ and $(0.9834, 0.3378)$ in $e_4$ and $e_5$ respectively. By scarifying a little bit MTA (around 0.01 in both cases), we can significantly reduce the ASR (around 0.1 in $e_4$ and 0.4 in $e_5$).

### F.6 Defense against adaptive malicious rate attack

We focus on an adaptive attacker strategy where the attack knows the time windows when the shadow learning framework conducts outlier detection and changes the number of malicious participating clients in each round accordingly. We assume that for every $r_0$ rounds, the attackers have the backdoor budget $B = \alpha n_\mathcal{C} r_0$, i.e., the adversary cannot corrupt more than $\alpha n_\mathcal{C} r_0$ participating clients every $r_0$ rounds, where $n_\mathcal{C}$ denotes the number of participating clients in each round.

In our experiments, we set $\alpha = 0.3$, $n_\mathcal{C} = 50$, $r_0 = 150$, and suppose the attacker has the knowledge that the outlier detection is conducted in a 30-round time window. In the time window, the attack corrupts $\alpha_{max}$-fraction participating clients, where $\alpha_{max} \geq \alpha$, intending to corrupt the learned SPECTRE filter. In the future 120 rounds, the malicious rate will drop to $\alpha_{min} = \frac{5}{4}\alpha - \frac{1}{4}\alpha_{max}$ due to unlimited backdoor budget. We vary $\alpha_{max}$ from 0.3 to 1 and show the ASR under shadow learning framework in Figure 9.

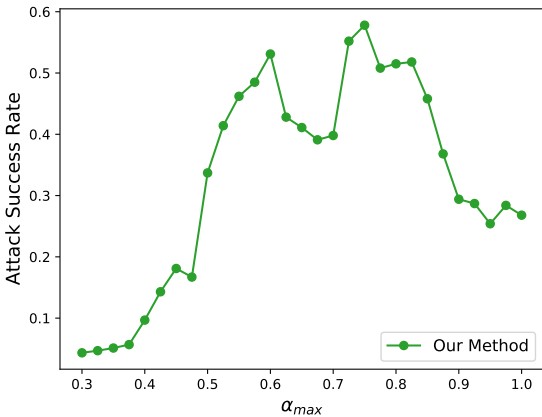

Figure 9: ASR of adaptive malicious rate attack with $\alpha = 0.3$

As observed in Figure 9, the ASR is smaller than 0.05 when $\alpha_{max} < 0.4$, indicating the robustness of shadow learning framework to the adaptive malicious rate attack with moderate $\alpha_{max}$. With the increase of $\alpha_{max}$, the SPECTRE filter will be corrupted and perform worse, and therefore, the ASR will increase when $\alpha_{max} < 0.6$. However, when $\alpha_{max}$ is large enough, $\alpha_{min}$ drops significantly such that there are no sufficient malicious clients to further corrupt the model even with more corrupted filter. Therefore, the ASR then drops as the $\alpha_{max}$ grows.

### F.7 Aggregation with noise

Our approach does not allow for secure aggregation, which may introduce privacy concerns. To this end, we consider the setting where all gradients and representations are corrupted with i.i.d. Gaussian noise $\mathcal{N}(0, \sigma^2)$, which is added to each coordinate before being uploaded from each client to the server. This is similar to Differentially-Private Stochastic Gradient Descent (DP-SGD). We vary the variance $\sigma^2$ from 0.001 to 1, and analyze the performance of our method in terms of (MTA, ASR) pairs under the homogeneous EMNIST dataset in Table 14.

Table 14: (MTA, ASR) Pair of Our Method under Different Noise Level

| $\alpha$ | $\sigma^2 = 0$ | $\sigma^2 = 0.001$ | $\sigma^2 = 0.01$ | $\sigma^2 = 0.1$ | $\sigma^2 = 1$ |
|---|---|---|---|---|---|
| 0.15 | (0.997, 0.007) | (0.998, 0.003) | (0.994, 0.003) | (0.983, 0.000) | (0.873, 0.000) |
| 0.25 | (0.998, 0.010) | (0.996, 0.013) | (0.995, 0.007) | (0.983, 0.000) | (0.863, 0.000) |
| 0.35 | (0.998, 0.031) | (0.998, 0.031) | (0.994, 0.027) | (0.980, 0.000) | (0.852, 0.000) |
| 0.45 | (0.997, 0.050) | (0.996, 0.050) | (0.993, 0.042) | (0.980, 0.017) | (0.867, 0.000) |

We can observe that as the noise level increases, both the MTA and ASR decrease. The intuition is that with large noise, the learning rate of the backdoor samples decreases, which makes the early-stopping framework more effective. However, the accuracy on main tasks also drops due to the noise. When $\sigma^2 = 0.1$, the ASR is 0 with $\alpha \leq 0.35$, while the MTA drops from around 0.997 to 0.98. With $\sigma^2 = 0.01$, the added noise can reduce the ASR while keeping MTA around 0.995.

## F.8 Sensitivity analysis of the filtering algorithm hyperparameters

For the filtering algorithm we adopt, there are three hyperparameters: malicious rate upper bound $\bar{\alpha}$, dimensionality reduction parameter $k$, and the QUE parameter $\beta$.

For $\bar{\alpha}$, we set the malicious rate $\alpha$ as its upper bound, i.e., $\alpha = \bar{\alpha}$ in most experiments. We also study the case where the malicious rate can be larger than its upper bound $\bar{\alpha}$ in Figure 9. For other hyperparameters, we set $k = 32$ and $\beta = 4$ in our experiments. We conduct the sensitivity analysis under CIFAR-100 for $k$ and $\beta$ in Table 15 and 16. We set the malicious rate as $\alpha = 0.15$.

Table 15: Sensitivity analysis of dimensionality reduction parameter $k$

| $k$ | 8 | 16 | 32 | 64 |
|---|---|---|---|---|
| ASR | 0.084 | 0.079 | 0.077 | 0.083 |
| MTA | 0.861 | 0.866 | 0.864 | 0.859 |

Table 16: Sensitivity analysis of dimensionality reduction parameter $\beta$

| $\beta$ | 2 | 4 | 6 | 8 |
|---|---|---|---|---|
| ASR | 0.081 | 0.077 | 0.082 | 0.082 |
| MTA | 0.867 | 0.864 | 0.859 | 0.853 |

We can observe that different values of hyperparameters $k$ and $\beta$ do not significantly affect the performance of our defense method.

