# OpenReview forum: "Towards a Defense Against Federated Backdoor Attacks Under Continuous Training"
_TMLR — Accepted by TMLR_

### Review · Reviewer_kR1q · 2023-04-17

**Summary Of Contributions:**

This paper proposes a defense method against backdoor attacks in federated learning. It trains an additional model, called a shadow model, other than the common global model (called the backbone model) at the server side. The shadow model is obtained by training on filtered updates that are cleaned by existing techniques. This paper assumes that the target label is known to the defender. During inference, it first uses the backbone model to make the prediction. If the predicted label is the target label, it then uses the shadow model to carry out another round of prediction, which will be used as the final output. The evaluation is conducted on four datasets. Compared to the baselines, the proposed method has better defense results.


**Audience:**

Yes

**Claims And Evidence:**

No

**Requested Changes:**

1. Evaluate the method on ImageNet or similar datasets with hundreds or thousands classes while the target label is unknown
2. Evaluate the scenario where the attacker injects the number of backdoors as the number of the classes
3. Conduct the experiment on real adaptive attacks targeting the filtering algorithm that this paper is based on
4. Discuss and compare with a highly related work

**Strengths And Weaknesses:**

Strengths

1. Defense against backdoor attacks in federated learning is an important research problem.
2. It is interesting to train a shadow model as a backup for correct prediction.

Weaknesses

1. The proposed method is based on the assumption that the defender knows the target label of the backdoor attack. This is a very unrealistic assumption. Even though the paper conducts the experiments to demonstrate that the proposed defense also works if the defender only knows the target label is within a set of labels (but not all of them). However, there are a few other problems.

   - Firstly, the proposed method needs to train a shadow model for each label, which is computationally infeasible. For ImageNet with 1000 classes, this method has to train and store 1000 models. It requires massive computation and storage resources.
   - Secondly, the attacker can inject multiple backdoors targeting different labels in the model. The proposed method can hardly work in such a scenario.

2. The effectiveness of this paper heavily depends on the existing filter algorithms. Those algorithms can be evaded by adaptive attacks. Although this paper evaluates adaptive attacks, none of the experiments considers the actual adaptive scenario, where the attacker particularly aims to evade the filter algorithm. The experiment in Appendix F.7 does not evaluate this. The L_ano loss is only for "the difference between the representations of backdoor samples and benign samples with target label". However, this is not the actual method used by those algorithms. Most of those filter algorithms aim to differentiate the clean and malicious updates. The adaptive attack shall try to evade this part, making malicious updates indistinguishable from clean ones.

3. The statement "there is no solution today that can defend against backdoor attacks in an FL setting with continuous training" is not true. An existing work [1] specifically targets the continuous training scenario and it can effectively remove backdoors in federated learning. This highly related work was not discussed and compared in the submission.


[1] Zhang, Kaiyuan, et al. "FLIP: A Provable Defense Framework for Backdoor Mitigation in Federated Learning." ICLR

---

> ### Author Response · Authors · 2023-05-02
> **Rebuttal (Q1)**
>
> **Q1: The proposed method is based on the assumption that the defender knows the target label of the backdoor attack. This is a very unrealistic assumption. Even though the paper conducts the experiments to demonstrate that the proposed defense also works if the defender only knows the target label is within a set of labels (but not all of them). However, there are a few other problems. Firstly, the proposed method needs to train a shadow model for each label, which is computationally infeasible. For ImageNet with 1000 classes, this method has to train and store 1000 models. It requires massive computation and storage resources. Secondly, the attacker can inject multiple backdoors targeting different labels in the model. The proposed method can hardly work in such a scenario.**
>
> We want to clarify that the experiments shown in Fig. 4 also demonstrate that the defense works for unknown target label attacks (For EMNIST dataset, when $|S_l|=10$, i.e., the target label is unknown to the defender, the attack success rate is $0.06$). For the assumption that the defender knows the target label or the label set that contains the target, we agree that it may not be feasible in every scenario, but this is reasonable when attackers have a clear motivation for backdooring (e.g., in spam detection, attackers are financially incentivized to label spam as ham).
>
> We also want to clarify that our algorithm only needs to train the shadow model for labels that could potentially be the target label. Our algorithm is computationally efficient if the potential target label set is small. Besides, for each shadow model, they just need to be trained for $R$ rounds ($R=50$ in our experiments), and then we can determine the actual target label and stop the other shadow models. This could also significantly reduce the computational cost. In the following experiments, we set $R=20$.
>
> To further demonstrate the performance of our algorithm on unknown target label defense, we set malicious rate $\alpha=0.15$ and conduct experiments on CIFAR-100 and Tiny-ImageNet datasets (with 100 and 200 classes respectively). The attack success rate (ASR) and main task accuracy (MTA) are shown as follows.
>
> |			|ASR		|MTA|
> |---|---|---|
> |CIFAR-100		|0.099		|0.862|
> |Tiny-ImageNet		|0.092		|0.801|
> ||
>
> The results show that our algorithm still works on datasets with hundreds of labels while the target label is unknown.
>
> To demonstrate that our algorithm works for multiple target label attacks, we conduct experiments on Tiny-ImageNet, and vary the number of backdoors injected. We set the malicious rate $\alpha=0.15$ and assume the target label is unknown. The maximal attack success rate among all target labels (Max-ASR) and main task accuracy (MTA) is shown as follows
>
> |#Backdoor|	Max-ASR|	MTA|
> |---|---|---|
> |1		|0.092		|0.801|
> |50		|0.107		|0.794|
> |100		|0.125		|0.813|
> |150		|0.122		|0.783|
> |200		|0.150		|Nan|
> ||
>
> We can observe that with the number of backdoors increasing, Max-ASR increases slightly from 0.092 to 0.150 while MTA keeps roughly the same, indicating that our algorithm can defend against attacks with multiple backdoors.
>
> We will add those results in our main paper.

---

> > ### Author Response · Authors · 2023-05-02
> > **Rebuttal (Q2, Q3)**
> >
> > **Q2: The effectiveness of this paper heavily depends on the existing filter algorithms. Those algorithms can be evaded by adaptive attacks. Although this paper evaluates adaptive attacks, none of the experiments considers the actual adaptive scenario, where the attacker particularly aims to evade the filter algorithm. The experiment in Appendix F.7 does not evaluate this. The $L_{ano}$ loss is only for "the difference between the representations of backdoor samples and benign samples with target label". However, this is not the actual method used by those algorithms. Most of those filter algorithms aim to differentiate the clean and malicious updates. The adaptive attack shall try to evade this part, making malicious updates indistinguishable from clean ones.**
> >
> > We want to clarify that the filter algorithm R-SPECTRE our framework currently adopts aims to differentiate the representation of backdoored and benign samples with target labels. Therefore, the adaptive attack shown in F.7 explicitly tries to evade R-SPECTRE by making malicious representations indistinguishable from benign ones. We currently choose R-SPECTRE as our filter algorithm because it has the slowest backdoor leakage (as shown in Fig. 1(a)).
> >
> > However, we agree that most filter algorithms aim to differentiate the clean and malicious updates (i.e., G-SPECTRE). We conduct experiments with adaptive attacks trying to masquerade malicious updates as  benign ones as follows. We adopt G-SPECTRE as the filter algorithm, and set the loss function as $L_{model} = \gamma L_{class} + (1-\gamma) L_{ano}$, where $L_{ano}$ accounts for the difference between clean and malicious updates. We set the malicious rate $\alpha=0.15$ and vary the value of $\gamma$ under CIFAR-100.
> >
> > |$\gamma$|	ASR|
> > |---|---|
> > |0.4		|0.101|
> > |0.6		|0.096|
> > |0.8		|0.087|
> > |1		|0.083|
> > ||
> >
> > The results indicate that the adaptive attack still cannot succeed under our defense with filter algorithms that differentiate the clean and malicious updates.
> >
> > **Q3: The statement "there is no solution today that can defend against backdoor attacks in an FL setting with continuous training" is not true. An existing work [1] specifically targets the continuous training scenario and it can effectively remove backdoors in federated learning. This highly related work was not discussed and compared in the submission.**
> >
> > **[1] Zhang, Kaiyuan, et al. "FLIP: A Provable Defense Framework for Backdoor Mitigation in Federated Learning." ICLR**
> >
> > Thank you for informing us of this work, we will update our paper to include this reference! This paper considers continuous backdoor attack in the federated setting (attack happens at least 60 rounds in their experiments), but not the continuous training setting (attack happens at least 1,200 rounds in our experiments). For this reason, we expect that it will experience backdoor leakage, as with the other baselines we tried. We tried to run experiments comparing our method (shadow learning) to FLIP, but were unable to find available code for FLIP. The github repository for the paper (https://github.com/KaiyuanZh/FLIP) only includes a README.

---

### Review · Reviewer_qGb3 · 2023-04-18

**Summary Of Contributions:**

The authors observed that different defenses against backdoors in federated learning suffer from a backdoor leakage, so that, if the model is trained for a large number of rounds, the attack success rate can increase significantly rendering the defenses useless. This can be particularly harmful for machine learning models under continuous training. Based on this observation, the authors propose a novel method to defend against backdoors in federated learning under continuous training using shadow learning, with a combination of two models, one trained using the entire (tainted) datasets provided by the participants, and another, aware of the target classes for the backdoor attack, using robust aggregation techniques, such as, for example, SPECTRE. The authors show that the combination of these two models allow to reduce the attack success rate against backdoor attacks, where the attackers just can manipulate the training datasets before training (i.e. it does not support model poisoning attacks). The authors also provide a mechanism to detect the target classes without previous knowledge.

**Audience:**

Yes

**Claims And Evidence:**

No

**Requested Changes:**

The paper makes an interesting point about the backdoor leakage of many defenses against backdoor attacks in federated learning that can be a problem in continuous training scenarios. The method proposed by the authors is interesting and promising. However, I think that the experimental evaluation is not convincing to support the claims made in the paper and validate the benefits of the proposed approach.

Thus, the main comparison in Section 5.1 includes a scenario where we do not have a continuous training setting. The authors used a static dataset for each participant, using IID partitions of the training set across the different clients. On the other side, the authors just reported the attack success rate and the accuracy and convergence of the different approaches is not shown. In this sense, it would be more appropriate to simulate a continuous training scenario and plot, both the attack success rate and the models’ accuracy (on benign data points) across the different training rounds. Apart from convergence, this analysis is interesting to observe what happens when the training datasets change during the continuous training of the models. It would also be more beneficial to do this assuming non-IID settings. In the paper, only a case with EMNIST is considered under non-IID settings, which is a bit limiting to extract useful conclusions.
The analysis of the defense without knowing the training label (Section 5.3) looks also shallow. I think that the comparison of both the cases where the defender is or not aware of the target label should be considered throughout all the experiments. On the other side, it is unclear how the parameters used for the filtering are set and a sensitivity analysis should be conducted to analyze the impact of this filter in the performance of the models and the attack success rate.

There is some abuse of the appendices. Some of the results reported there would fit better in the main paper. For instance, Section 5.5 consists only on two lines of text pointing to the appendices. In this sense, the analysis against adaptive attacks and against different triggers is of enough relevance to show in the main sections of the paper (the authors spend more pages for the appendices than for the main paper).

Apart from the experiments, the threat model (adversarial model as named in the paper) is a bit unclear. I think that the authors should clarify this and follow a similar threat model as in SPECTRE (Hayase et al. 2021). Compared to other papers, the threat model in this case is more restrictive than for some of the competing methods used in the experimental evaluation. Thus, in this paper, only poisoning attacks, where the adversary can only inject malicious points, but cannot manipulate the training procedure directly, performing, for example, model poisoning attacks. Actually, in this sense, the comparison in Section 5.1 is not really fair, as the other competing methods are design to resist attacks in more challenging settings. It is fine for me that the authors just focus on data poisoning attacks, but the position of the paper and the threat model should be clearer about this. For instance, in Algorithm 1 (lines 4 and 10), the malicious clients can lie on the accuracy.

Some other minor comments or clarifications:
+ In page 4: how is the backdoored dataset constructed in the aggregator?
+ Page 5: the backbone model “is used to learn parameters of a filter that can filter out poisoned model updates.” Can the authors clarify this?
+ In Section 5.1: how many training iterations? 1,200 or 12,000?
+ Experimental settings: how do the authors generate heterogeneous partitions of the datasets for the different clients? It’s not clear from the information in Appendix E.


**Strengths And Weaknesses:**

Strengths:
+ The observation about the backdoor leakage for many defenses against backdoors in federated learning is interesting, and makes the case for continuous training. It shows the need for tackling this problem with a different angle.
+ The proposed method using a combination of two models, with one of them, aware of the target labels, looks promising, especially, as the authors included a method to defend against backdoors even if the target labels are unknown.
+ Although some of the assumptions are a bit restrictive, the theoretical analysis included in Section 4 is nice to support the benefits of the proposed method (although perhaps, more discussion on the implications of the theoretical results would be beneficial).

Weaknesses:
-	The threat model and the positioning with respect to other defenses in the state of the art is not very clear. Perhaps, the authors can consider the threat model for SPECTRE as a reference for this paper.
-	The experimental evaluation is not very convincing and/or comprehensive (see comments in the requested changes).
-	There is some abuse on the use of the appendix. Given that there is no limitation on the number of pages, the paper would be more readable is some of the information given in the appendices is moved to the main paper.

---

> ### Author Response · Authors · 2023-05-02
> **Rebuttal (Q1, Q2, Q3)**
>
> **Q1: The main comparison in Section 5.1 includes a scenario where we do not have a continuous training setting. The authors used a static dataset for each participant, using IID partitions of the training set across the different clients. On the other side, the authors just reported the attack success rate and the accuracy and convergence of the different approaches is not shown. In this sense, it would be more appropriate to simulate a continuous training scenario and plot, both the attack success rate and the models' accuracy \(on benign data points\) across the different training rounds. Apart from convergence, this analysis is interesting to observe what happens when the training datasets change during the continuous training of the models.**
>
> Thank you for the suggestion. We would like to clarify that the scenario in Section 5.1 is the continuous training setting without data distribution shifts. Fig.1(a) and Fig.5 are the plots that show the attack success rate (ASR) across different training rounds under this setting when malicious rate $\alpha = 0.03$. For the main task accuracy (MTA), both convergence rate and the converged value are roughly the same over different approaches. We will add this plot to the main paper.
>
> Under the static settings, with different malicious rates $\alpha$, the main task accuracy for our method is alway above $0.99, 0.94, 0.85, 0.7$ for EMNIST, CIFAR-10, CIFAR-100 and Tiny-ImageNet respectively. We will further emphasize this in our paper. The comparisons of convergence rate are similar to Fig.1(a) and Fig.5. We will add more plots in the final version of the paper.
>
> For the dynamic settings where data distribution changes over time (Section 5.2, Appendix F.5), Figure 3, 7, 8 show the ASR and MTA across different training rounds at the same plot. Each curve in these plots can be regarded as the combination of ASR curve and MTA curve for a certain defense method. Based on these plots, we can more directly compare the convergence rate of ASR and MTA, which is critical to evaluate the efficiency of early-stopping (early-stopping is efficient if the curve achieves top-left, i.e., high MTA and ASR ).
>
> **Q2: It would also be more beneficial to do this assuming non-IID settings. In the paper, only a case with EMNIST is considered under non-IID settings, which is a bit limiting to extract useful conclusions.**
>
> Thank you for the suggestion. We conduct more experiments in the non-IID settings with dataset CIFAR-100 and Tiny-ImageNet. We set malicious rate $\alpha=0.15$ and vary the
> heterogeneity level $h$ (described in Appendix F.4). The ASR and MTA of our method is shown as follows.
>
> For CIFAR-100:
> |heterogeneity level $h$	|0	|0.2	|0.4	|0.6	|0.8	|1.0|
> |---|---|---|---|---|---|---|
> |ASR				|0.077	|0.079	|0.086	|0.091	|0.094	|0.103|
> |MTA				|0.864	|0.859	|0.861	|0.858	|0.851	|0.853|
> ||
>
> For Tiny-ImageNet:
> |heterogeneity level $h$|	0	|0.2	|0.4	|0.6	|0.8	|1.0|
> |---|---|---|---|---|---|---|
> |ASR				|0.072	|0.071	|0.078	|0.083	|0.089	|0.091|
> |MTA				|0.793	|0.767	|0.798	|0.802	|0.789	|0.794|
> ||
>
> For all baselines, the model is completely backdoored due to the backdoor leakage, and the ASR is always larger than $0.79$ and $0.60$ for CIFAR-100 and Tiny-ImageNet respectively.
>
> We will add those results and the corresponding plots to the final version of the paper.
>
> **Q3: The analysis of the defense without knowing the training label \(Section 5.3\) looks also shallow. I think that the comparison of both the cases where the defender is or not aware of the target label should be considered throughout all the experiments.**
>
> Thank you for the suggestion. To further demonstrate the performance of our algorithm on unknown target label defense, we set malicious rate $\alpha=0.15$ and conduct experiments on CIFAR-100 and Tiny-ImageNet datasets (with 100 and 200 classes respectively). The ASR and MTA are shown as follows.
>
> |			|ASR		|MTA|
> |---|---|---|
> |CIFAR-100		|0.099		|0.862|
> |Tiny-ImageNet		|0.092		|0.801|
> ||
>
> Again, for all baselines, the model is completely backdoored due to the backdoor leakage, and the ASR is always larger than $0.75$ and $0.60$ for CIFAR-100 and Tiny-ImageNet respectively.
>
> We will add those results to the final version of the paper.

---

> > ### Author Response · Authors · 2023-05-02
> > **Rebuttal (Q4, Q5, Q6)**
> >
> > **Q4: On the other side, it is unclear how the parameters used for the filtering are set and a sensitivity analysis should be conducted to analyze the impact of this filter in the performance of the models and the attack success rate.**
> >
> > For the filter algorithm, there are three hyperparameters: malicious rate upper bound $\bar{\alpha}$, dimensionality reduction parameter $k$, and the QUE parameter $\beta$.
> >
> > For $\bar{\alpha}$, we set the malicious rate $\alpha$ as its upper bound, i.e., $\alpha=\bar{\alpha}$ in most experiments. We also study the case where the malicious rate can be larger than its upper bound $\bar{\alpha}$ in Fig.9. For other hyperparameters, we set $k=32$ and $\beta = 4$. We conduct the sensitivity analysis under CIFAR-100 for $k$ and $\beta$ as follows. We set the malicious rate $\alpha=0.15$.
> >
> > |k	|ASR		|MTA|
> > |---|---|---|
> > |8	|0.084		|0.861|
> > |16	|0.079		|0.866|
> > |32	|0.077		|0.864|
> > |64	|0.083		|0.859|
> > ||
> >
> > |$\beta$|	ASR		|MTA|
> > |---|---|---|
> > |2	|0.081		|0.867|
> > |4	|0.077		|0.864|
> > |6	|0.082		|0.859|
> > |8	|0.082		|0.853|
> > ||
> >
> > We can observe that different values of hyperparameters $k$ and $\beta$ do not affect the performance of our defense method a lot.
> >
> > **Q5: There is some abuse of the appendices. Some of the results reported there would fit better in the main paper. For instance, Section 5.5 consists only on two lines of text pointing to the appendices. In this sense, the analysis against adaptive attacks and against different triggers is of enough relevance to show in the main sections of the paper \(the authors spend more pages for the appendices than for the main paper\).**
> >
> > We are sorry that we put so much content into the appendices. We will move the analysis against adaptive attacks, different triggers and more results for dynamic settings to the main paper in the final version.
> >
> > **Q6: Apart from the experiments, the threat model (adversarial model as named in the paper) is a bit unclear. I think that the authors should clarify this and follow a similar threat model as in SPECTRE \(Hayase et al. 2021\). Compared to other papers, the threat model in this case is more restrictive than for some of the competing methods used in the experimental evaluation. Thus, in this paper, only poisoning attacks, where the adversary can only inject malicious points, but cannot manipulate the training procedure directly, performing, for example, model poisoning attacks. Actually, in this sense, the comparison in Section 5.1 is not really fair, as the other competing methods are design to resist attacks in more challenging settings. It is fine for me that the authors just focus on data poisoning attacks, but the position of the paper and the threat model should be clearer about this. For instance, in Algorithm 1 (lines 4 and 10), the malicious clients can lie on the accuracy.**
> >
> > Thank you for pointing this out. For our threat model, we assume the adversary corrupts clients independently in each round, and they cannot corrupt more than fraction $\alpha$ of the participating clients in that round. In each round, malicious clients can contribute whatever data they want. However, we assume that they otherwise follow the protocol. This is the same threat model as in SPECTRE (Hayase et al. 2021) and RFA (Pillutla et al. 2019). In Appendix F.6, we consider a stronger threat model where the adversary does not have the upper bound on malicious rate $\alpha$, but instead have a backdoor budget $B$ every $r$ rounds, i.e., the adversary cannot corrupt more than $B$ participating clients every $r$ rounds.
> >
> > We agree that our threat model does not allow model poisoning attacks, and will clarify the position of the paper and the threat model in the final version. However, we want to emphasize that even under our threat model, all the previous works will be completely backdoored due to the backdoor leakage.

---

> > > ### Author Response · Authors · 2023-05-02
> > > **Rebuttal (Q7, Q8, Q9, Q10)**
> > >
> > > **Q7: In page 4: how is the backdoored dataset constructed in the aggregator?**
> > >
> > > We would like to clarify that only the malicious local clients can construct the backdoored datasets by injecting backdoor triggers. The aggregator only maintains the backbone and shadow models. These models might be backdoored by aggregating the malicious model updates sent by the adversary. We will make this more clear in our paper.
> > >
> > > **Q8: Page 5: the backbone model “is used to learn parameters of a filter that can filter out poisoned model updates.” Can the authors clarify this?**
> > >
> > > We are sorry for the confusion. When the backbone model converges on the target label, each client locally averages the representations of samples with the target label, and this average is sent to the server for filtering. To get the filter, the server calls Algorithm 2 to get filter parameters (line 12 in Algorithm 1). These parameters are used in Algorithm 3 to remove malicious clients (line 15 in Algorithm 1).
> > >
> > > Since the representations can be taken from the penultimate layer of the backbone model, we say the backbone model “is used to learn parameters of a filter that can filter out poisoned model updates.”
> > >
> > > **Q9: In Section 5.1: how many training iterations? 1,200 or 12,000?**
> > >
> > > The number of training iterations in Section 5.1 is 1,200. Since all the models converge in fewer than 500 iterations under the static settings, we train the model for 1,200 rounds to model continuous training.
> > >
> > > **Q10: Experimental settings: how do the authors generate heterogeneous partitions of the datasets for the different clients? It’s not clear from the information in Appendix E.**
> > >
> > > We are sorry for the confusion. The heterogeneous partition is described in Appendix F.4. Specifically, h-fraction of the overall training images are shuffled and evenly partitioned to each client, and for the remaining (1 − h)-fraction samples, each client receives shuffled images from 4 randomly selected classes with $\lfloor 25(1 − h) \rfloor$ samples per class. We will include this information when we introduce heterogeneous experiments.

---

### Review · Reviewer_kLkR · 2023-04-19

**Summary Of Contributions:**

This paper focus on the defense against backdoor attacks in federated learning. In backdoor attacks, the training data is sourced from untrusted clients. In the federated learning scenario, defending backdoor attacks is more difficult since the server do not have access to raw client data. The authors propose shadow learning framework to defend against backdoor attacks in the FL setting under long-range training. They train two models in parallel, one of which is trained to make predictions for samples that are not in the target class, another of which is trained to make predictions for the backdoor target class. They claim to be the first work to protect against backdoor attacks in FL under continuous training.

**Audience:**

Yes

**Claims And Evidence:**

Yes

**Requested Changes:**

See the above questions.

**Strengths And Weaknesses:**

Strengths:
1. To my knowledge, this paper is the first work to address the backdoor problem in FL under continuous training, which is enlightening for further research.
2. They observe a phenomenon of backdoor leakage where most backdoor defenses lose efficacy over many training rounds.
3. This paper is well-written, with the majority of its sections being easily comprehensible.

Questions:
1. This paper focus on addressing the backdoor problem in FL under continuous training and emphasize that the continuous training is different from continual learning. Is the continuous training of importance in FL practice? Does other FL research consider the continuous training diagram or this paper propose it firstly? If so, why is the continuous training diagram necessary? The paper did not state this aspect clearly.
2. The experiments in Section 5.1 is performed under homogeneous and static clients, while the experiments under distribution drift is used for ablation study. Since the authors emphasize the continuous training diagram, the comparison between baselines and the proposed method under distribution drift is necessary.
3. The second constraint in Section 1: “FL models are typically trained continuously e.g., due to distribution drift” worth more discussion. In many real-world application and research, distribution drift or statistical heterogeneity is very important. I want to see more specific discussion on it.

---

> ### Author Response · Authors · 2023-05-02
> **Rebuttal**
>
> **Q1: This paper focus on addressing the backdoor problem in FL under continuous training and emphasize that the continuous training is different from continual learning. Is the continuous training of importance in FL practice? Does other FL research consider the continuous training diagram or this paper propose it firstly? If so, why is the continuous training diagram necessary? The paper did not state this aspect clearly.**
>
> Continuous training is one of the key properties of federated learning, and there are several works [1,2] focused on the issues raised by continuous training in FL, e.g., communication cost and algorithm design.
> We differentiate continuous training from continual learning since we mainly focus on the difficulties in backdoor defense raised by continuous training rather than try to solve the classical problems in continual learning (e.g., catastrophic forgetting), which we think is an orthogonal direction. We will make this more clear in the paper.
>
> [1] Yu, Bin, et al. "A survey on federated learning in data mining." Wiley Interdisciplinary Reviews: Data Mining and Knowledge Discovery 12.1 (2022): e1443.
>
> [2] Xu, Chenming, and Yunlong Mao. "An improved traffic congestion monitoring system based on federated learning." Information 11.7 (2020): 365.
>
> **Q2: The experiments in Section 5.1 is performed under homogeneous and static clients, while the experiments under distribution drift is used for ablation study. Since the authors emphasize the continuous training diagram, the comparison between baselines and the proposed method under distribution drift is necessary.**
>
> Thank you for your suggestion. Section 5.1 indicates that even without distribution drift, all baselines are completely backdoored due to backdoor leakage. For the dynamic settings where data distribution shifts over time, all baselines are still fully backdoored. Section 5.2 indicates that our method can successfully defend against backdoor attack under distribution drift, and all the proposed components are necessary. We will add the complete comparison between all baselines, Periodic R-SPECTRE, Shadow Network Prediction, and our method under distribution drift in the final version of the paper.
>
> **Q3: The second constraint in Section 1: “FL models are typically trained continuously e.g., due to distribution drift” worth more discussion. In many real-world application and research, distribution drift or statistical heterogeneity is very important. I want to see more specific discussion on it.**
>
> Distribution shift and temporal data heterogeneity exist widely in real-word applications, e.g., traffic congestion monitoring [1] and healthcare monitoring [2]. In order to adapt to changing data distributions, models should be trained continuously, both in the federated and central settings [3]. In some applications, enterprises may want to update their model to perform well on the current distribution, as well as on previously-seen distributions (i.e., they want to prevent catastrophic forgetting). However, in this paper, we consider a simpler setting in which the central party only wants to perform well on the currently-seen distribution. Even in this simpler setting, we show that existing approaches are unable to resist backdoor attacks. Hence, we view solving the current problem as a precursor to fully solving the backdoored continual learning problem in the federated setting.
>
> We will add more discussion on this point to the final version of our paper.
>
> [1] Xu, Chenming, and Yunlong Mao. "An improved traffic congestion monitoring system based on federated learning." Information 11.7 (2020): 365.
>
> [2] Brophy, Eoin, et al. "Estimation of continuous blood pressure from PPG via a federated learning approach." Sensors 21.18 (2021): 6311.
>
> [3] Hofer, Vera, and Georg Krempl. "Drift mining in data: A framework for addressing drift in classification." Computational Statistics & Data Analysis 57, no. 1 (2013): 377-391.

---

### Author Response · Authors · 2023-07-24
**Camera Ready Submission**

Dear Action Editor,

We thank you for your decision and suggestions. We have submitted the camera-ready paper with the required revision, i.e., discuss the expensive computation cost of checking every label and incorporate the reviewers' suggestions during the rebuttal. Please let us know if you would like to see any other changes. Thanks to you and to the reviewers for your constructive feedback!

---

### Decision · Action_Editors · 2023-06-06

**Recommendation:** Accept with minor revision

**Comment:**

Three expert reviewers reviewed this paper with ratings "Accept", "Leaning Accept", "Leaning Reject". After the rebuttal, authors addressed majority of the concerns; however, there are still outstanding issues to be fixed.

The paper can be accepted after authors make the following revision.

1- Authors should explicitly and clearly discuss the expensive computation cost of checking every label. This is raised by one of the reviewers and is an important point to emphasize.

2- Authors should incorporate the reviewers' suggestions including the results provided during the rebuttal.

**Audience:**

The paper is on a important subject that is certainly relevant to TMLR community.

**Claims And Evidence:**

The paper focuses on defending against backdoor attacks in federated learning (FL) where training data is sourced from untrusted clients. The authors propose training an extra shadow model to mitigate backdoor attacks in FL under continuous training, that is, they train two models simultaneously: one for samples not in the target class and another for the backdoor target class. The proposed approach is claimed to reduce the attack success rate against backdoor attacks, and protecting against backdoor attacks in FL.

Authors evaluate the proposed methodology on four datasets. They compare against baselines, and argue that the proposed method has better defense capabilities.